# BIGDML—Towards accurate quantum machine learning force fields for materials

Huziel E. Sauceda [1,2,3✉], Luis E. Gálvez-González[4], Stefan Chmiela [2,5], Lauro Oliver Paz-Borbón[6], Klaus-Robert Müller [2,5,7,8,9✉] & Alexandre Tkatchenko [10✉]

Machine-learning force fields (MLFF) should be accurate, computationally and data efficient, and applicable to molecules, materials, and interfaces thereof. Currently, MLFFs often introduce tradeoffs that restrict their practical applicability to small subsets of chemical space or require exhaustive datasets for training. Here, we introduce the Bravais-Inspired Gradient-Domain Machine Learning (BIGDML) approach and demonstrate its ability to construct reliable force fields using a training set with just 10–200 geometries for materials including pristine and defect-containing 2D and 3D semiconductors and metals, as well as chemisorbed and physisorbed atomic and molecular adsorbates on surfaces. The BIGDML model employs the full relevant symmetry group for a given material, does not assume artificial atom types or localization of atomic interactions and exhibits high data efficiency and state-of-the-art energy accuracies (errors substantially below 1 meV per atom) for an extended set of materials. Extensive path-integral molecular dynamics carried out with BIGDML models demonstrate the counterintuitive localization of benzene–graphene dynamics induced by nuclear quantum effects and their strong contributions to the hydrogen diffusion coefficient in a Pd crystal for a wide range of temperatures.

[1] Departamento de Materia Condensada, Instituto de Física, Universidad Nacional Autónoma de México, Cd. de México C.P. 04510, Mexico. [2] Machine Learning Group, Technische Universität Berlin, 10587 Berlin, Germany. [3] BASLEARN - TU Berlin/BASF Joint Lab for Machine Learning, Technische Universität Berlin, 10587 Berlin, Germany. [4] Programa de Doctorado en Ciencias (Física), División de Ciencias Exactas y Naturales, Universidad de Sonora, Blvd. Luis Encinas & Rosales, Hermosillo, C.P. 83000, Mexico. [5] BIFOLD – Berlin Institute for the Foundations of Learning and Data, Berlin, Germany. [6] Departamento de Física Química, Instituto de Física, Universidad Nacional Autónoma de México, Cd. de México C.P. 04510, Mexico. [7] Google Research, Brain team, Berlin, Germany. [8] Department of Artificial Intelligence, Korea University, Anam-dong, Seongbuk-gu 02841 Seoul, Korea. [9] Max Planck Institute for Informatics, Stuhlsatzenhausweg 66123 Saarbrücken, Germany. [10] Department of Physics and Materials Science, University of Luxembourg, L-1511 Luxembourg City, Luxembourg. ✉email: huziel.sauceda@fisica.unam.mx; klaus-robert.mueller@tu-berlin.de; alexandre.tkatchenko@uni.lu

The development and implementation of accurate and efficient machine learning force fields (MLFF) is transforming atomistic simulations throughout the fields of physics[1–5], chemistry[6–14], biology[15,16], and materials science[17–22]. The application of MLFFs have enabled a wealth of novel discoveries and quantum-mechanical insights into atomic-scale mechanisms in molecules[3,6,22–25] and materials[2,4,26–28].

A major hurdle in the development of MLFFs is to optimize the conflicting requirements of ab initio accuracy, computational speed and data efficiency, as well as universal applicability to increasingly larger chemical spaces[29]. In practice, all existing MLFFs introduce tradeoffs that restrict their accuracy, efficiency, or applicability. In the domain of materials modeling, all MLFFs known to the authors employ the so-called locality approximation, i.e. the global problem of predicting the total energy of a many-body condensed-matter system is approximated by its partitioning into localized atomic contributions. The locality approximation has been rather successful for capturing local chemical degrees of freedom, as demonstrated in a wide number of applications[30–34]. However, we emphasize that the locality assumption disregards non-local interactions and its validity can only be truly assessed by comparison to experimental observables or explicit ab initio dynamics. This fact restricts truly predictive MLFF simulations of realistic materials, whose properties are often determined by a complex interplay between local chemical bonds and a multitude of non-local interactions.

The chemical space of materials is exceedingly diverse if we count all possible compositions and configurations of a given number of chemical elements. For example, an accurate MLFF reconstruction of the potential-energy surface (PES) of elemental bulk materials to meV/atom accuracy often requires many thousands of configurations for training[21,30,35–38]. The MLFF errors also increase at least by an order of magnitude when including defects or surfaces[32,35].

Heteroatomic materials and interfaces between molecules and materials would require substantially more training data for creating predictive MLFFs and accuracies much better than 1 meV/atom, eventually making the modeling of such materials intractable. In addition, there is a strong desire to go beyond traditional density-functional theory (DFT) reference data in the field of atomistic materials modeling[39–41]. Beyond-DFT methods can only be realistically applied to compute dozens or hundreds of geometries, making the construction of beyond-DFT MLFFs impractical.

To address these challenges, in this work we introduce a Bravais-Inspired Gradient Domain Machine Learning (BIGDML) model for periodic materials that is accurate, data efficient, and computationally inexpensive at the same time. The BIGDML model extends the applicability domain of the Symmetric Gradient-Domain Machine Learning (sGDML) framework[23,42,43] to include periodic systems with supercells containing up to roughly 200 atoms. The BIGDML model employs a global representation of the full system, i.e. treating the supercell as a whole instead of a collection of atoms. This avoids the uncontrollable locality approximation, but also restricts the maximum number of atoms in the unit cell. To extend the applicability of BIGDML to much larger unit cells will require the development of a global multiscale representation. An additional advantage of a global representation is that cross-correlations between forces on different atomic species are dealt with rigorously. Specifically, MLFFs based on the locality approximation construct separate models for each atom type. In contrast, the BIGDML model employs a global force covariance, allowing many-body correlations between atomic forces in a given supercell structure and capturing relevant interatomic interactions at different spatial scales. Similarly to the sGDML model,

another key advantage of the BIGDML model is the usage of physical constraints (energy conservation) and all relevant physical symmetries of periodic systems, including the full translation and Bravais symmetry groups. As a consequence, BIGDML models achieve meV/atom accuracy already for 10–200 training points, surpassing state-of-the-art atom-based models by 1-2 orders of magnitude. This result underlines once again the importance of including prior knowledge, including physical laws and symmetries, into ML models. Clearly, what is known does not need to be learned from data—in effect the data manifold has been reduced in its complexity (see e.g.[23,24,34,44–47]). It is important to mention that describing materials with several hundreds of atoms, having an exceedingly large number of symmetries, or transferring models between different systems still remain challenging tasks for the BIGDML model. Nevertheless, these technical issues could be addressed by utilising multiscale and composite approaches as well as iterative numerical kernel solvers (see the Discussion section for an extended discussion).

Altogether, the BIGDML framework opens the possibility to accurately reconstruct the PES of complex periodic materials with unprecedented accuracy at very low computational cost. In addition, the BIGDML model can be straightforwardly implemented as an ML engine in any periodic DFT code, and used as a molecular dynamics driver after being trained on just a handful of geometries.

## Results

The BIGDML framework relies on two advances: (i) a global atomistic representation with periodic boundary conditions (PBC), (ii) the use of the full translation and Bravais symmetry group for a given material.

**PBC-preserving representation**. To avoid localization of interatomic interactions and artificial (from the electronic perspective) atom-type assignment, we use an efficient global representation with PBC. Following the sGDML approach for molecules[23,42], we take the atomistic Coulomb matrix (CM)[48] as a starting representation. When used with sGDML, the CM has been proven to be a robust, accurate, and efficient representation[23,42,49].

Here, we introduce a generalization of the molecular CM descriptor to represent periodic materials, $\mathcal{D}^{(\text{PBC})}$. In order to construct the Coulomb matrix for extended systems, we first enforce the PBC using the minimal-image convention (MIC)[50,51]:

$$\mathcal{D}_{ij}^{(\text{PBC})} = \begin{cases} \frac{1}{|\mathbf{r}_{ij} - \mathbf{A}\text{mod}(\mathbf{A}^{-1}\mathbf{r}_{ij})|} & \text{if } i \neq j \\ 0 & \text{if } i = j \end{cases} \quad (1)$$

where $r_{ij} = r_i - r_j$ is the difference between two atomic coordinates $i$ and $j$, and A is the matrix defined by the supercell translation vectors as columns. Figure 1A left shows the Coulomb matrix descriptor when considering only the supercell structure with no PBC, which means that the ML model considers the system as a finite "molecule". The right side of Fig. 1A shows the descriptor with the PBC enforced (Eq. (1)), having now the correct periodic structure.

Many widely used periodic global representations already exist, for example CM-inspired global descriptors such as the Ewald-sum, or extended Coulomb-like and sine matrices[52]. In the cases of the extended Coulomb-like and Ewald matrices, these representations account the contribution of the same atom iteratively by considering its multiple periodic images, which is computationally demanding and algebraically involved. From these global periodic representations[52], only the sine matrix avoids using redundant information, since it just depends on the atomic positions in a single unit cell. Nevertheless, this is only a good representation for studying the crystal structure of materials

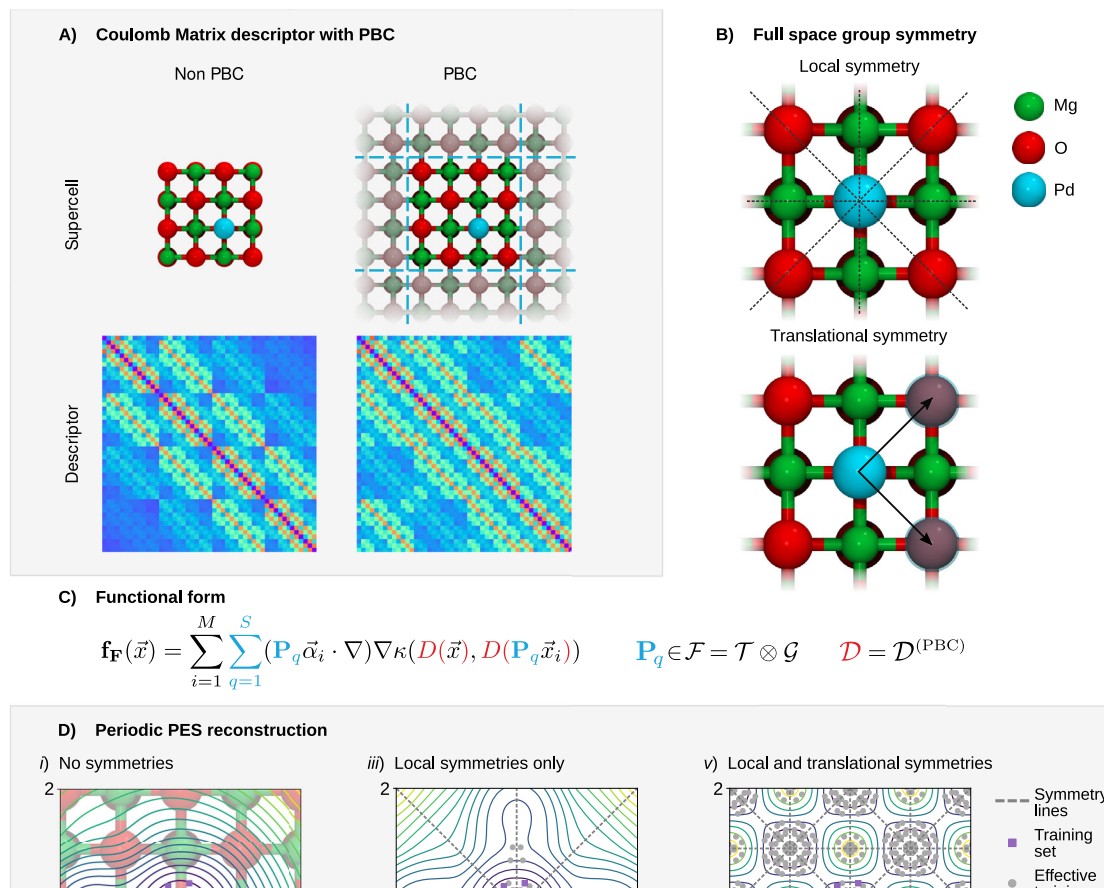

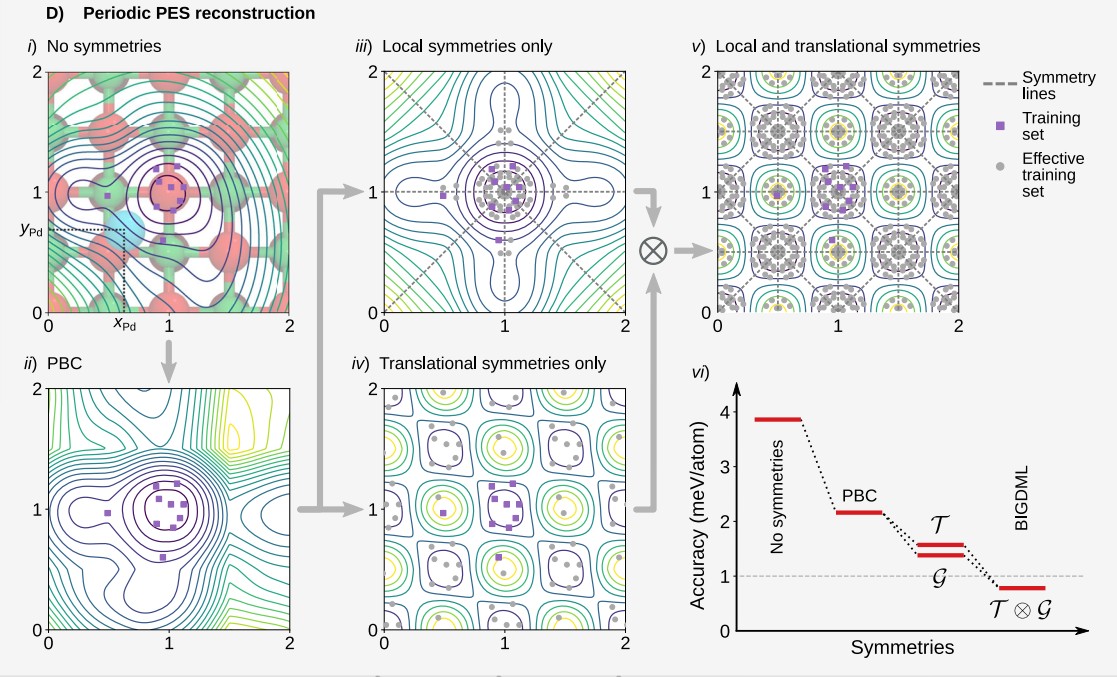

**Fig. 1 The BIGDML model. A** Coulomb Matrix representation for non-periodic (left) and periodic (right) supercell (2 × 2 × 3L) of Pd$_1$/MgO(100).
**B** Description of the local symmetries (i.e. Bravais group $\mathcal{G}$ or point group of the unit cell) and the translation symmetries of the unit cell $\mathcal{T}$. **C** Analytical form of the sGDML predictor, where the explicit usage of the full symmetry group of the supercell $\mathcal{F}$ (in blue) and the Coulomb matrix PBC descriptor (in red). **D** Systematic symmetrization of the PES. The axis in the PES are the $x$ and $y$ coordinates of the Pd atom (purple squares) in units of the lattice constant of MgO. Models: (i) Pure GDML, (ii) GDML +$\mathcal{D}^{(PBC)}$, (iii) sGDML +$\mathcal{D}^{(PBC)}$(s =$\mathcal{G}$), (iv) sGDML+$\mathcal{D}^{(PBC)}$(s =$\mathcal{T}$), and (v) BIGDML. The gray arrows indicate the order in which each symmetry was applied, and the panel shows its effect on the PES. (vi) displays the incremental accuracy upon addition of each symmetry for the Pd$_1$/MgO (100) system. All models used for this comparison were trained on 50 data samples.

at equilibrium, given that its main feature (i.e. projection of the full supercell to a single unit cell) cannot provide an accurate measure of two atoms in different cells. Additionally, in the case where the supercell of the system is taken as a "unit cell" in the sine matrix, the obtained descriptor is essentially local and unable to capture long-range interatomic correlations within the super-cell. Our choice of CM with PBC enforced using MIC is one of

the simplest and efficient choices, which also turns out to be exceptionally accurate and data-efficient, as will be shown below.

As an alternative to the global approach, many local materials' representations have been developed. Among those representa-tions, there are numerous descriptors based on atomic local environments, for example atom-density representations[53–56], partial radial-distribution functions[57], FCHL descriptor[58],

rotationally-invariant internal representation[59], many-body vector interaction[60] and moment tensor potentials[18]. In all these cases, the PBC can be naturally incorporated by using the MIC, as it has been done for mechanistic force fields. These local representations in principle aim at the construction of transferable interatomic MLFFs, as done by GAP/SOAP framework[55] which is the basis of a series of high-quality chemical bonding potentials for phosphorus[32], carbon[35], and silicon[21]. However, the intrinsic cutoff radius in these descriptors limits the extent of atomic environments, neglecting the ubiquitous long-range interactions and correlations between different atomic species. Here, by using a $\mathcal{D}^{(PBC)}$ global descriptor, we avoid the need of fine-tuning representation hyperparameters while preserving high accuracy in the description of the many possible configuration states of a material.

**Translation symmetries and the Bravais' group**. The full symmetry group $\mathcal{F}$ of a crystal is given by the semidirect product of translation symmetries $\mathcal{T}$ and the rotation and reflection symmetries of the Bravais lattice $\mathcal{G}$ (Bravais' group): $\mathcal{F} = \mathcal{T} \otimes \mathcal{G}$[61] (See Fig. 1B). This is a general result, meaning that it applies to any periodic system of dimension $d$, $\mathcal{F}^{(d)} = \mathcal{T}^{(d)} \otimes \mathcal{G}^{(d)}$. In practice, the translation group $\mathcal{T}$ is constructed by the set of translations of the Bravais cell that span the supercell using the primitive translation vectors as a basis, while the Bravais' group $\mathcal{G}$ is the symmetry point group of the unit cell. In order to illustrate these concepts, as an example, let us consider a graphene ($d = 2$) supercell of size $5 \times 5$. Its full symmetry group is $\mathcal{T}^{(2)}_{5 \times 5} \otimes \mathcal{G}^{(2)} = T^{(2)}_{5 \times 5} \otimes D_{6h}$ and contains 300 symmetry elements. Further important materials with ample symmetries are surfaces and interfaces. Analogous to molecules possessing internal rotors, molecules interacting with a surface are another case of a fluxional system. For example, benzene adsorbed on graphene has a full fluxional symmetry group defined by the direct product of graphene's full symmetry group and benzene's molecular point group, $[T^{(2)}_{5 \times 5} \otimes D_{6h}]_{graphene} \otimes [D_{6h}]_{benzene}$, which contains 3600 symmetry elements. Such a large number of symmetries reduces considerably the region of configuration space needed to be sampled to reconstruct the full PES and consequently generate MLFF models with high data efficiency. The presented arguments generalize to other materials, such as molecular crystals, rigid bulk materials, porous materials, and hybrid organic-inorganic materials, e.g. perovskites.

**The BIGDML model**. The construction of a BIGDML model consists in combining a global PBC-descriptor and the full symmetry group of the system in the gradient-domain machine learning framework (See Fig. 1), which leads to a robust and highly data efficient MLFF, capable of reaching state-of-the-art accuracy using only a few dozens of training points. We would like to stress here that such unprecedented data efficiency opens up many opportunities to study advanced materials using high levels of electronic-structure theory, such as sophisticated DFT approximations or even coupled-cluster theory[62].

In a nutshell, the periodic global supercell descriptor and symmetries presented in the previous sections are combined with the sGDML framework to create the BIGDML predictor displayed in Fig. 1C. To illustrate the effects of the symmetries in the PES reconstruction process for the atom–surface $Pd_1$/MgO system, Fig. 1D presents a diagram where the different core elements of the BIGDML model are systematically included and the resulting (learned) PES is displayed. In this figure, the shown PES corresponds to the energy surface experienced by a Pd atom. The panel (i) displays the reconstructed energy surface with no

symmetries, where the training samples are the purple squares and represent the position of the Pd atom. In panel (ii) the PBC are enforced by the periodic descriptor (Eq. (1)), and then this is combined with the use of the point group of the unit cell in panel (iii) and with translation symmetries in panel (iv). From the last two panels, we can see the characteristic contribution of each symmetry group, $\mathcal{G}$ symmetrizes the local PES by adding effective training samples (shown as grey circles) while $\mathcal{T}$ delocalises the effective sampling over the whole supercell. Then, by considering the full symmetry group $\mathcal{F}$, in panel (v) we arrive to the PES reconstructed by the BIGDML model where the effective training data symmetrically span the whole supercell. The panels (i) to (v) show the increasing symmetrization of the PES, but also illustrate the accuracy gain at each stage. The prediction accuracy plot shown in panel (vi) clearly shows the important impact of each symmetry group in generating accurate and robust BIGDML models. It is important to highlight that the achieved accuracy is a combination of several complementary contributions. The Coulomb matrix analytical form provides the correct description of long-range interactions, the Matern kernel provides a basis function that correctly describes the tails of the contributions of each training point (See Methods section), and the full crystal symmetry group correctly enforces the symmetries in the predictor function. This can be seen in (Fig. 1-D-vi), where the accuracy of the BIGDML model increases consistently as each element is added.

**Prediction performance of BIGDML for different materials**. The BIGDML model can be applied to accurately reproduce atomic forces and total energy of bulk materials, surfaces, and interfaces. To illustrate the applicability of BIGDML, in this section we have selected representative systems that cover the broad spectrum of materials, and study the prediction accuracy of our MLFFs as judged by the learning curves (test error as a function of the number of data points used for training). The considered systems include bulk materials (graphene as a representative 2D material, 3D metallic and semiconducting solids), surfaces (Pd absorbed on MgO surface), and van der Waals bonded molecules on surfaces (benzene adsorbed on graphene), as well as a bulk material with interstitial defects (hydrogen in palladium). Additionally, we analyse the case of the $CsPbBr_3$ perovskite to test the performance of the model for larger supercells. For a detailed description of the database generation and the levels of theory, as well as the parameters of the simulations and software packages employed, we refer the reader to the Methods section.

**Bulk materials: graphene as a representative 2D material**. Graphene is a well-characterized layered material that continues to exhibit many remarkable properties despite being thoroughly studied[36,63,64]. Hence, developing accurate and widely applicable force fields for graphene and its derivatives is an active research area. Recently, Rowe et al.[36] presented a comprehensive comparison of existing hand-crafted force fields and a Gaussian-process approximated potential (GAP) using the Smooth Overlap of Atomic Positions (SOAP) local descriptor. The GAP/SOAP approach was shown to generalize much better than mechanistic carbon FFs. In Fig. 2 we show the learning curves of the BIGDML model for $5 \times 5$ supercell of graphene, showing that only 10 geometries (data samples) are needed to match the best-performing method to date ($\approx 25$ meV $Å^{-1}$ in force RMSE)[36]. The performance and data efficiency of BIGDML is remarkable, given that it uses less than 1% of the amount of data employed by atom-based local descriptors. More importantly, by increasing the number of data samples used for training to 100, we reach a

## A) Energy prediction

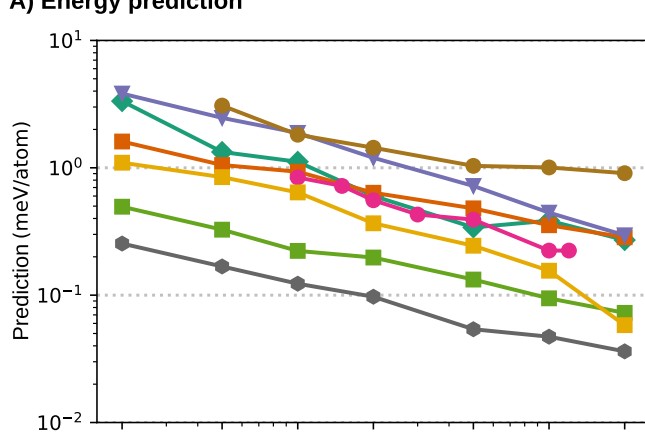

## B) Force prediction

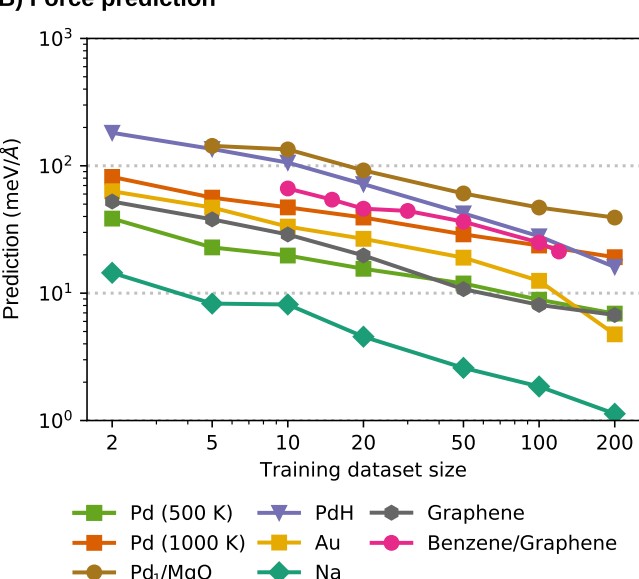

Pd (500 K) — PdH — Graphene

Pd (1000 K) — Au — Benzene/Graphene

Pd$_1$/MgO — Na

**Fig. 2 Learning curves for different materials. A** Energy and (**B**) Forces. 3D bulk materials: Pd(FCC), Na(BCC), and Au(FCC). 2D material: Graphene. Interstitial in materials: H in a supercell of Pd. Chemisorption of atom at a surface: Single Pd atom adsorbed on a MgO (100) surface. Van der Waals interactions: Benzene molecule adsorbed on graphene. Their respective full (fluxional) symmetry group, supercell dimensions and reference levels of theory used in each case are presented in Methods section. The reported values are the mean absolute errors (MAE). See Supplementary Fig. 1 for additional information on the learning curves.

### A) Energy prediction          ### B) Atomic force prediction

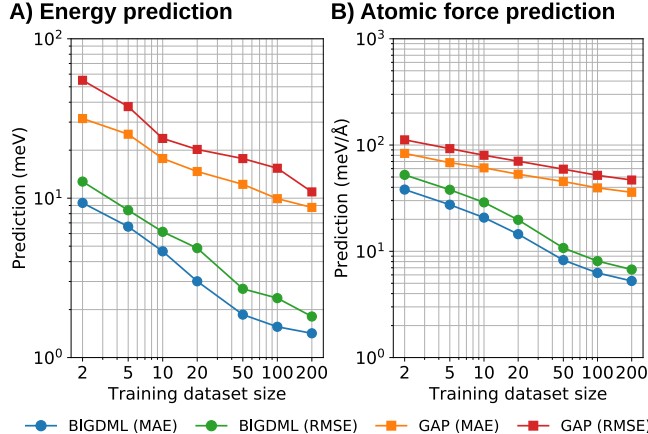

— BIGDML (MAE)  — BIGDML (RMSE)  — GAP (MAE)  — GAP (RMSE)

**Fig. 3 BIGDML and GAP performance comparison for graphene.** Comparison of the (**A**) energy and (**B**) forces learning curves, both trained using the same datasets.

and Na[BCC]. Figure 2 shows the learning curve for these three structures with a supercell of $3 \times 3 \times 3$ and symmetry groups $T^{(3)}_{3 \times 3 \times 3} \times O_h$. An accuracy of $\approx 10$ meV $Å^{-1}$ for a monoatomic metal material can be achieved using approximately 70 samples in the case of Pd (only 10,000 atomic forces), which is only a fraction of the data (less than 1%) required by other models to obtain the same accuracy[38].

**Bulk materials: large perovskite supercell**. In order to test the usability of the model for larger supercells, now we consider the case of the CsPbBr$_3$ perovskite, which contains 160 atoms per supercell. This all-inorganic perovskite is a system of great interest in photovoltaics given its stability under highly humid environments, hence it is a potential candidate for reliable solar cells. These materials are known to have high fluxionality which drive the system to visit multiple local minima at finite temperature. Therefore, allowing long molecular dynamic trajectories via ML models helps to collect better sampling statistics, and hence more robust physical observables. In Fig. 4, energy and forces learning curves are shown. Despite the larger supecell, already at 100 training samples we reach the $\approx 1$ meV/atom energy accuracy and a force error of 50.7 meV/Å. Furthermore, we found that when training on 1000 samples, BIGDML manages to achieve energy and force accuracy of $\approx 0.1$ meV/atom and $\approx 2.6$ meV/Å, respectively.

The obtained accuracy demonstrates that the BIGDML model can also achieve high fidelity in the reconstruction of the PES for large multi-element supercells with rich fluxional dynamics, as it is the case of perovskite materials.

**Surfaces: atom chemisorbed at a surface –Pd$_1$/MgO**. One of the main challenges of constructing MLFFs on local atomic environments is that such representations can fail to capture subtle local changes with global implications. For example, when describing a surface or an interface, atoms of the same element are described by the same atomic embedding function which in order to encode the many possible neighbourhoods (atoms in deeper layers, atoms close to the surface of the material) requires large amounts of training data. This eventually leads to degradation of MLFF performance, a problem that could become practically intractable for local MLFFs when dealing with molecule-surface interactions. These limitations can be addressed in local models but at the cost of higher complexity models and manual tuning of hyperparameters, hence losing the key advantages of MLFFs. In this section, we show that the BIGDML

generalization error of $\approx 1$ meV (0.02 meV/atom) in energies and $\approx 6$ meV $Å^{-1}$ for forces. To our knowledge, such accuracies have not been obtained in the field of MLFFs for extended materials. In order to put our results into context of state-of-the-art MLFFs, in Fig. 3 we show the learning curves comparing GAP/SOAP and BIGDML for graphene (See Supplementary Fig. 2 for an extended comparison using different materials). Given the same data for training, BIGDML achieves an improvement of a factor of 10 in accuracy, both for the total energy and atomic forces. The same conclusions hold for other systems studied in this work, as shown in the Supplementary Figs. 1 and 2.

**Bulk materials: the case of cubic crystals**. In the case of 3D materials, we apply our model to monoatomic metallic materials covering common cubic crystal structures: Pd[FCC], Au[FCC]

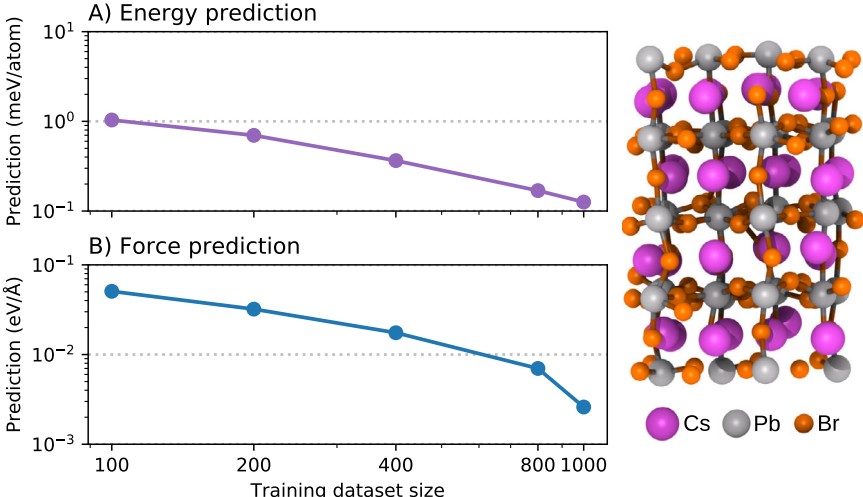

**Fig. 4 Model performance on large supercells. A** Energy and (**B**) force learning curves of the BIGDML model for CsPbBr$_3$ perovskite. The right panel displays the supercell structure containing 160 atoms.

method does not have such limitations by studying two representative systems: chemisorbed Pd/MgO-surface and physisorbed benzene/graphene.

In recent years, it has been shown that single-atom catalysts (SACs) can offer superior catalytic performance compared to clusters and nanoparticles[65–67]. These heterogeneous catalysts consist of isolated metal atoms supported on a range of substrates, such as metal oxides, metal surfaces or carbon-based materials. As a showcase, here we use a single Pd atom supported on a pristine MgO (100) surface. The considered supercell consists of a $2 \times 2$ slab of MgO(100) with 3 layers, where the lowest layer is kept fixed, and a single Pd atom is deposited on the surface.

The full symmetry group for this system is $T^{(2)}_{2 \times 2} \otimes C_{4v}$ with 64 elements. The learning curve (see Fig. 2) shows that only 200 samples are needed to reach energy and force accuracy values of $\approx 34$ meV ($\approx 0.7$ meV/atom) and $\approx 30$ meV Å$^{-1}$, respectively. Similarly as in the case of learning force fields for molecules in the gas phase, the target error is always relative to the relevant dynamics of the system and its energetics[23,42,43]. In this context, the Pd atom is chemisorbed at an oxygen site and the lowest energetic barrier that the Pd atom experiences is of 450 meV, thus our error is $\approx 6\%$ of this value. In Fig. 5 we show the minimum-energy barrier (MEB) of Pd atom displacing from one minimum to another on the MgO surface computed by the nudged elastic band (NEB) method (See Methods section for details). It must be noted that the Pd atom never crossed this barrier during the MD simulation used to generate the reference dataset, as displayed by the purple lines in Fig. 5 indicating the distribution of the Pd atom location in the training dataset. Hence, even though the model did not have information regarding the saddle point, the energetic barrier was nevertheless correctly modeled by BIGDML by incorporating translational and Bravais symmetries.

**Surfaces: molecule physisorbed at a surface –Benzene/graphene**. A highly active field of research in materials science concerns the interaction between molecules and surfaces, due to its fundamental and technological relevance. From the modeling point of view, describing non-covalent interactions within the framework of DFT remains a competitive research area given its intricacies, which has led to very accurate dispersion interaction methods[68–72]. Nevertheless, most of the studies about these systems focus on global optimizations or short MD simulations. Here, we demonstrate the applicability of BIGDML by learning

the molecular force field of the benzene molecule interacting with graphene.

The full symmetry group of the benzene/graphene system is $T^{(2)}_{5 \times 5} \otimes C^{(Graphene)}_{6v} \otimes C^{(Benzene)}_{6v}$, which has a total of 3600 elements. This large number of symmetries greatly reduces the configurational space sampling requirements to reconstruct its PES, as can be seen from the learning curve shown in Fig. 2 where the energy error quickly drops below $\approx 43$ meV (1 kcal mol$^{-1}$) training only on 10 datapoints and $\approx 21$ meV with 30 training datapoints. For this system, the energy generalisation accuracy starts to saturate at 0.18 meV/atom when training on 100 configurations. Achieving such high generalization accuracy using only a handful of training data points for such a complex system convincingly illustrates the high potential of the BIGDML model, since it suddenly opens the possibility of performing predictive simulations for a wide variety of systems where only static DFT calculations are available so far.

The systems discussed in this section offer a general picture of the broad diversity of extended materials that the BIGDML model can describe with high data efficiency and unprecedented accuracy. In particular, the applications here introduced provide a range of families of systems and materials that can be described by the model. For example, it is to be expected that, in general, given the performance of the trained models for Na, Au, and Pd, mono-atomic materials with cubic unit cells will be accurately described by the BIGDML. On the other hand, the accurate description of the CsPbBr$_3$ perovskite material shows that the model can handle and accurately learn large multi-element and fluxional materials. Then, a similar performance is expected when applied to a family of materials with similar structural characteristics. In the same order of ideas, the performance and applicability of the BIGDML model to molecules interacting with 2D materials is demonstrated with the benzene/graphene system, given that similarly complex dispersion interactions are to be expected.

**Validation of BIGDML models for materials properties**. In the previous section, we demonstrated the prediction capabilities of the BIGDML method using statistical accuracy measures. Now, we assess the predictive power of BIGDML models in terms of predicting physical properties of materials. In this section, we first perform a thorough test for ML models by assessing the phonon spectra of 2D graphene and 3D bulk materials. Then, we proceed to test the performance beyond the harmonic approximation by

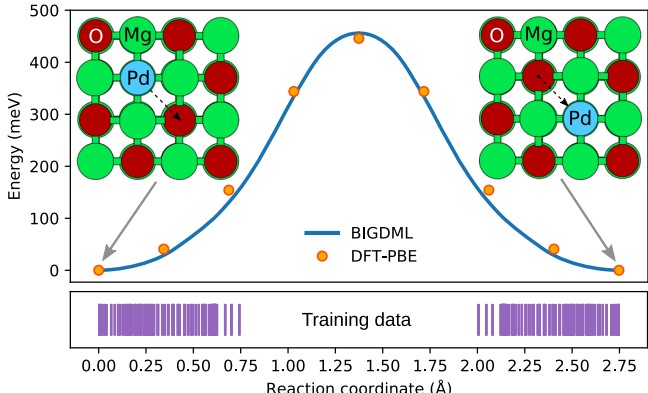

**Fig. 5 Model performance on unseen energy barriers.** Comparison of the minimum-energy path between neighbouring oxygen-sites for a single Pd supported on a MgO (100) surface using BIGDML (continuous line) and the reference method, DFT/PBE (circles). The purple lines indicate the location of the Pd atom in the training dataset.

carrying out molecular dynamics simulations and comparing observables against explicit DFT calculations. All simulations performed in this section were done using the best trained models displayed in the learning curves (See Fig. 2 and "Methods" section).

**Phonon spectra.** A common challenging test to assess force fields (machine learned[20,35,36] as well as conventional FFs[73–75]) is the phonon dispersion curves and phonon density of states, since they give a clear view of (i) the proper symmetrization of the FF and (ii) the correct description of the elastic properties of the material in the harmonic approximation. The main challenge for FFs is describing both collective low-frequency phonon modes and the local high-frequency ones with equal accuracy. In Fig. 6 we show the comparison of the BIGDML and DFT generated phonon bands displaying a perfect match, showing a RMSE phonon errors across the Brillouin zone of 0.85 meV for Graphene, 0.35 meV for Na, and 0.38 meV for Pd. These values are comparable to those reported in literature using MLFFs trained on thousands of configurations and hand-crafted datasets[76], while in our case we only require less than 100 randomly selected training points. Such accuracy originates from the use of a global representation for the supercell which captures local and non-local interactions with high fidelity, a feature that is crucial in describing vibrational properties.

Now, we proceed to a more challenging physical test, which is the prediction of properties at finite temperature, where also the anharmonic parts of the PES are important.

**Molecular dynamics simulations: graphene.** Simulations of graphene at finite temperature using an accurate description of the interatomic forces is a highly relevant topic given the plethora of applications of this material. In particular, a necessary contribution to its realistic description is the inclusion of nuclear quantum effects (NQE). For example, the experimental free energy barrier for the permeability of graphene-based membranes to thermal protons can only be correctly described by including the NQE of the carbon atoms[77,78]. In order to corroborate that our graphene BIGDML model is giving the correct physical delocalization of the nuclei, we performed path-integral molecular dynamics (PIMD) simulations at 300 K for a 5 × 5 supercell. In Fig. 7 we compare the distribution of first neighbor interatomic distance $r_{CC}$ between classical MD (blue) and PIMD (orange), results showing that the fluctuations in $r_{CC}$ double its value when

considering NQE. These findings are in excellent agreement with explicit first-principles PIMD simulations in the literature[78].

As an additional robustness test, we have performed extended classical MD simulations at various temperatures using the EAM force field[79] and a BIGDML model trained on this level of theory, obtaining a perfect match between these two different methods. This further validates the predictive power of our methodology even at long time scales. These results are shown in the Supplementary Fig. 5.

Up to this point, we have performed simulations to validate our models under different conditions. In the next section, we perform predictive simulations, which highlight the potential of BIGDML for novel applications, including unexpected NQE-driven localization of benzene/graphene dynamics and the diffusion of interstitial hydrogen in bulk palladium.

**Validation of BIGDML in dynamical simulations of materials.** Benzene/graphene. The interaction between different molecules and graphene has been extensively studied, given the potential applications of molecule/graphene systems as electrical and optical materials and even as candidates for drug delivery systems[80–90]. Of particular interest is the understanding of the effective binding strength and structural fluctuations of adsorbed molecules at finite temperature, which requires long time-scale molecular dynamics simulations, unaffordable when using explicit ab initio calculations. Here we will demonstrate that BIGDML models can be used for studying explicit long-time dynamics of realistic systems such as benzene (Bz) adsorbed on graphene with accurate and converged quantum treatment of both electrons and nuclei (See Fig. 8A). The Bz/graphene system has three minima that resemble those of the benzene dimer: the π − π stacking (parallel-displaced) structure as global minimum and two local minima corresponding to parallel and T-shaped configurations, as displayed in Fig. 8-B[3] along with the corresponding structural parameters and adsorption energies computed at the PBE+MBD level of theory[69,70,91]. The calculated adsorption energy for the global minimum is in a very good agreement with experimental measurements of 500 ± 80 meV[92].

An extensive amount of studies exist on the implications of NQE on properties of molecules and materials at finite temperature[93,94], however much less is known about the implications of NQE for non-covalent van der Waals (vdW) interactions[3,95]. In the particular case of Bz/graphene, considering the translational symmetries of the PES experienced by the Bz molecule as well as thermal fluctuations and its many degrees of freedom, it is to be expected that the Bz dynamics will be highly delocalized. Nonetheless, it was recently reported that the inclusion of NQE in a molecular dimer can considerably enhance intermolecular vdW interactions[3]. However, the adsorption/binding energy ratio between Bz/graphene and Bz/Bz system is $E_{ads}^{Bz/graphene}/E_{int}^{Bz/Bz} \approx 4$, therefore it is not clear how NQEs will affect such strongly interacting vdW systems.

In order to assess the role of temperature and NQE for Bz/graphene, in Fig. 8C we present the results obtained from classical MD and PIMD simulations at 300K using a BIGDML FF trained at the PBE+MBD level of theory. At this temperature, the benzene molecule tends to mostly populate configurations at an angle of ≈ 10° relative to the graphene normal vector in both cases (see Fig. 8A). Nevertheless, classical MD simulations explore substantially wider regions of the PES, reaching angles of up to 80°, close to the T-shaped minimum. In contrast, PIMD simulations yield a localized sampling of θ with a maximum angle of ≈ 30°. To understand the origin of this localization, we have systematically increased the "quantumness" of the system by raising the number of beads in the PIMD simulations to converge

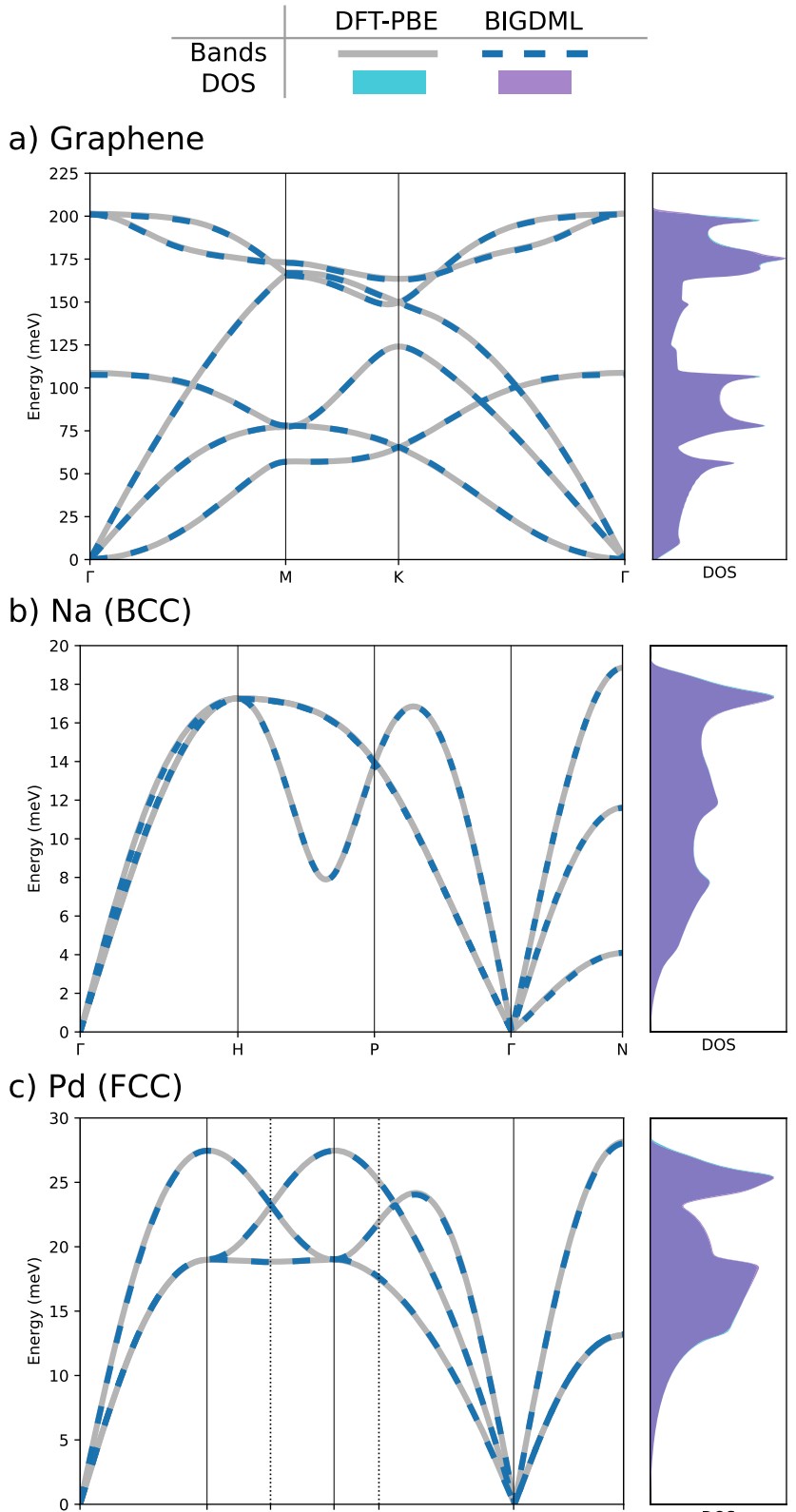

**Fig. 6 Model performance on phonon bands reconstruction.** Phonon spectra (left panels) and vibrational densities of states (right panels) for (**a**) graphene, (**b**) bulk sodium, and (**c**) bulk palladium along the high-symmetry paths in their respective Brillouin zones. The dashed line represents BIGDML and the continuous line the reference DFT-PBE level of theory. The differences between DFT and BIGDML are visually imperceptible.

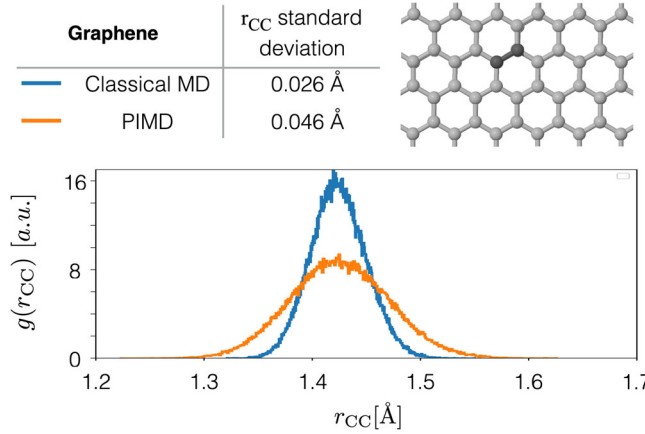

| Graphene | $r_{CC}$ standard deviation |
|---|---|
| Classical MD | 0.026 Å |
| PIMD | 0.046 Å |

**Fig. 7 Nuclear quantum effects in graphene.** Radial distribution function for first-neighbours in graphene (top panel). Classical (MD, in blue) and path integral molecular dynamics (PIMD, in orange) simulations describing the graphene system at room temperature generated by the BIGDML molecular force field. A Gaussian function fitting gives std. of 0.026 Å and 0.046 Å for classical MD and PIMD, respectively (table in top panel).

towards the exact treatment of NQE. This approach provides concrete evidence of the progressive localization of the benzene normal orientation as the NQE increase (see Supplementary Fig. 3). The physical origin of this phenomenon is the NQE-induced interatomic bond dilation, where the zero-point energy generated by NQE drive the system beyond the harmonic oscillation regime. The intramolecular delocalization produces effective molecular volume dilation and increases the average polarizability of benzene and graphene rings, akin to a recent analysis of non-covalent interactions between molecular dimers upon constraining their center of mass[3]. In contrast, in this work no constraints were imposed on the Bz/graphene system, suggesting that the Bz molecule localization on graphene should be observable in experiments. In order to further rationalize the NQE-induced stabilization of vdW interactions, we have computed the vdW interaction energy as a function of compression/dilation of the Bz molecule on graphene and found a linear dependence between dilation and vdW interaction (see Supplementary Fig. 4). This analysis fully supports our hypothesis of NQE-induced stabilization and dynamical localization.

The rather fundamental nature of the underlying physical phenomenon of NQE-induced stabilization suggests that many polarizable molecules interacting with surfaces will exhibit a similar dynamical localization effect. It is worth mentioning that a thorough analysis of the Bz/graphene system demands extensive simulations, which are now made accessible due to the computational efficiency and accuracy of the BIGDML model. Our modeling could also be applied to larger molecules with peculiar behavior under applied external forces[96].

**Hydrogen interstitial in bulk palladium.** Hydrogen has become a promising alternative to fossil fuels as a cleaner energy source. Nevertheless, finding a safe, economical and high-energy-density hydrogen storage medium remains a challenge[97]. One of the proposed methods is to store hydrogen in interstitial sites of the crystal lattices of bulk metals[97–99]. Among these metals, palladium has been widely researched as a candidate, since it can absorb large quantities of hydrogen in a reversible manner[98].

Characterizing the diffusion of hydrogen in the crystal lattices at different temperatures is crucial to assess their performance as storage materials. Hence, in this section we study a system consisting of a hydrogen atom interstitial in bulk palladium with a cubic supercell containing 32 Pd atoms with full symmetry

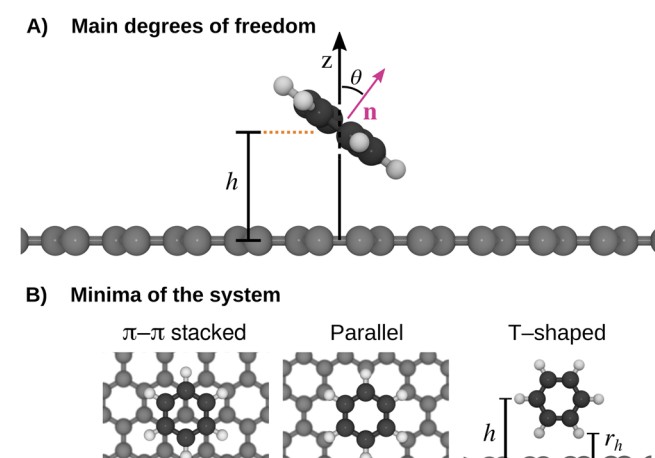

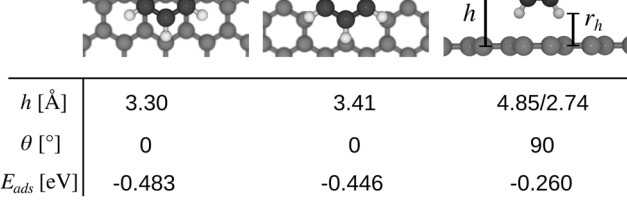

| | $\pi$–$\pi$ stacked | Parallel | T–shaped |
|---|---|---|---|
| $h$ [Å] | 3.30 | 3.41 | 4.85/2.74 |
| $\theta$ [°] | 0 | 0 | 90 |
| $E_{ads}$ [eV] | -0.483 | -0.446 | -0.260 |

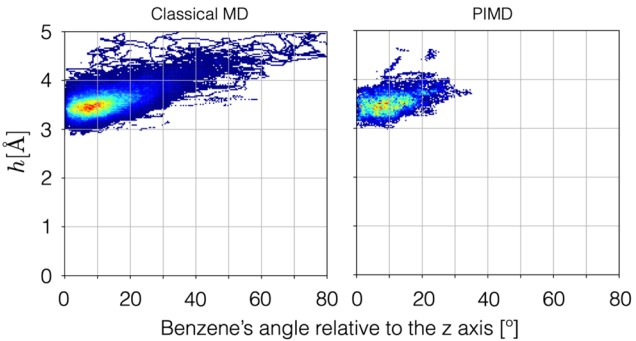

**Fig. 8 Dynamical strengthening of non-covalent interactions.** Benzene/graphene. **A** Depiction of the system with its two main degrees of freedom: (1) The angle between the normal vector defined by the benzene ring $\hat{n}$ and the normal to the graphene plane ($\hat{z}$). (2) The relative distance between the benzene center of mass and the graphene, $h$. **B** The three minima of the systems and its defining characteristic parameters: $h$, $\theta$, and its adsorption energy in $E_{ads}$. The $E_{ads}$ energies were computed using $E_{ads} = E_{benzene/graphene} - (E_{benzene} + E_{graphene})$. **C** Classical molecular dynamics (MD) and path integral molecular dynamics (PIMD) simulations describing the dynamics of benzene molecule interacting with graphene at room temperature described by the BIGDML molecular force field. Plots displaying projections of the dynamics (classical MD and PIMD) to the two main degrees of freedom of the system: $h$, $\theta = \arccos(\hat{z} \cdot \hat{n})$.

group $T^{(3)}_{2\times2\times2} \otimes O_h$, and described at the DFT-PBE level of theory (See Methods section for more details). The BIGDML learning curve for this system in presented in Fig. 2. Within the FCC lattice there are two possible cavities for hydrogen atoms storage: the octahedral (O-sites) and the tetrahedral (T-sites) cavities (See Fig. 9A-top), where the O-site is the global minimum[98] and it is separated from the T-site by an energetic barrier of $\approx 160$ meV as shown in Fig. 9A-bottom. Additionally, from this figure, we can see the excellent agreement between BIGDML model and the reference DFT calculations.

Kimizuka et al.[98] reported a study based on transition state theory (TST) suggesting that not only the inclusion of the NQE has indeed a strong effect on the H-atom diffusion, but also they reported that NQE hinder the migration from O-site to T-site. In order to elucidate realistic dynamics of the H atom in the metal

**Fig. 9 Interstitial hydrogen diffusivity in bulk Pd. A** (Top) Minima of the trajectory used to calculate the minimum-energy path (MEP) for the diffusion of H in FCC Pd. (Bottom) MEP as calculated using BIGDML (continuous line) and DFT-PBE (circles) between adjacent O- and T-sites. **B** Three-dimensional plots of the probability distribution of the H atom in the O-site at different temperatures. **C** Diffusivity of H in bulk Pd as a function of the temperature. The cyan squares and red diamonds represent the values calculated running classical MD and PIMD using the BIGDML model, respectively. The lines are the Arrhenius fit to the data from MD@BIGDML (cyan, dashed), PIMD@BIGDML (red, solid) and TST-PIMD by Kimizuka et al. (gray, dash-dotted)[98]. Experimental data were taken from refs. [117] (blue circles)[118], (orange triangles) and[119] (green crosses).

lattice and the impact of the NQE without relying on approximations such as TST, we performed direct classical MD and PIMD simulations at different temperatures (from 100 K to 1000 K) (see "Methods" for more details). We first studied the NQE-induced statistical sampling of the hydrogen atom in each cavity as shown in Fig. 9B. In Supplementary Videos 1–4, we can see an animated version of this figure, where the shape of the sampled volume and how it changes as a function of the temperature and by the inclusion of the nuclear quantum effects is displayed. This helps us to visualize hydrogen dynamics in the temperature range from 100 to 300 K and to determine the shape of the cavity, which transforms from a cube to a much larger truncated octahedron as the temperature increases.

Then, from the generated (classical and quantum) trajectories we have estimated the diffusivity of the hydrogen atom as a function of the temperature, which are shown in Fig. 9C along with TST results and experimental data. Usually, this quantity is estimated by using transition rate theory, which only considers the energetics of the system (energy barrier and relative energy between adjacent states)[98]. A more robust methodology is to directly compute the diffusivity from the molecular dynamics results using the mean-square displacement analysis[100], an option which MLFFs make feasible due to the long-time simulations required while keeping ab initio accuracy. From these results, we observe an Arrhenius temperature dependence for the diffusivity ($D(T) = D_0 e^{-Q/kT}$) in both the classical MD and the PIMD cases, which is expected in the range of temperatures considered. While the TST-PIMD results by Kimizuka et al.[98] ($D_0 = 9.90 \times 10^{-7}$ m$^2$/s, $Q = 0.23$ eV) accurately reproduce the experimental activation barrier $Q$ for H in Pd, they considerably overestimate the value of the pre-exponential factor $D_0$. In our case, the diffusivity of H in Pd at lower temperatures is overestimated by classical MD ($D_0 = 0.95 \times 10^{-7}$ m$^2$/s, $Q = 0.151$ eV), but it is close to the reported values at high temperatures. Meanwhile, both the activation barrier ($Q = 0.231$ eV) and the pre-exponential factor ($D_0 = 2.70 \times 10^{-7}$ m$^2$/s) calculated using BIGDML@PIMD are in excellent agreement with the experimental data at all temperatures.

The results presented in this section demonstrate how BIGDML enables long PIMD simulations to obtain novel insights into dynamical behaviour of intricate materials containing vacancies or interstitial atoms.

## Discussion

In this work, we introduced the BIGDML approach—a MLFF for materials that is accurate, straightforward to construct, efficient in terms of learning on reference ab initio data, and computationally inexpensive to evaluate. The accuracy and efficiency of the BIGDML method stems from extending the sGDML framework for finite systems[23,42] by employing a global periodic descriptor and making usage of translational and Bravais symmetry groups for materials. The BIGDML approach enables carrying out extended dynamical simulations of materials, while correctly describing all relevant chemical and physical (long-range) interactions in periodic systems contained within the reference data. In principle, once high-level electronic structure force calculations for periodic systems (with CCSD(T) or Quantum Monte Carlo methods) become a reality[39–41], the BIGDML method would be an ideal tool to execute highly accurate dynamics of materials. We remark that the molecular sGDML approach has already fulfilled this long-standing goal for molecules with up to a few dozen atoms[3,23].

We have demonstrated the applicability and robustness of the BIGDML method by studying a wide variety of relevant materials and their static and dynamical properties, for example successfully assessing the performance of BIGDML models for physical observables in the harmonic and anharmonic regimes in the form

of phonon bands and molecular dynamics simulations. Furthermore, we carried out predictive simulations on interstitial hydrogen diffusion in bulk Pd, as well as accurately capturing intricate van der Waals forces and the dynamics of the interface formed by molecular benzene and 2D graphene layer.

From the practical perspective, the BIGDML approach represents an advantageous framework beyond its accuracy and data efficiency, given that the model generation is a straightforward process starting from the simplicity of database generation and its out-of-the-box training procedure[43]. From the deployment point of view, to illustrate the gain in computational speed, we remark that for benzene/graphene we gain a factor of 50,000 for computing atomic forces with BIGDML when compared to the PBE + MBD level of electronic-structure theory. This gain would further increase when using a higher level of quantum-mechanical methods for generating reference data.

Many powerful MLFFs for materials have been proposed, and some are already widely used for materials modelling[54,101]. In order to embed the BIGDML model into the current context of MLFFs for materials, it is convenient to address some of the limitations that current methodologies face, as well as to discuss goals to pursue with the next generation of MLFFs in materials science.

All current MLFFs for materials known to the authors employ the locality approximation, i.e. they build a model for an energy of an atom in a certain chemical environment, which is defined by a cutoff function. The typical employed cutoffs are of 3–8Å, being of a rather short range. Increasing the cutoff does not necessarily lead to a better model, because electronic interactions exhibit hard-to-learn multiscale structure[9]. From the practical point of view, trying to embed more information in the atomic representation by increasing the cutoff radius leads to other problems such as learning capacity limitations and, in the case of neural networks, their inherent difficulty to correctly represent and propagate multiscale interactions. In addition, different interaction scales are mutually coupled. An attractive feature of the locality approximation is that in principle, the short-range interactions are transferable to different systems. However, in practice this is not a general finding. For example, it was shown that a general-purpose GAP/SOAP MLFF for carbon[35] yields errors an order of magnitude higher in graphene compared to the same methodology trained specifically on graphene data[36]. In addition, local MLFFs typically decouple interaction potentials of different atoms by assigning atom types. For example, carbon in benzene and carbon in graphene could be treated as different atom types. Obviously, such decoupling makes the learning problem harder because more data is necessary to "restore the coupling" between different atomic species.

BIGDML provides a robust solution for both problems of localization by using a global descriptor with periodic boundary conditions. This allows BIGDML to capture interactions at all relevant length scales by virtue of coupling all atomic coordinates. However, such a prominent feature comes with a limitation, since transferablity between different systems is not easy to achieve. The current approach to achieve transferable MLFFs is the localization of interactions. Thereby, transferability remains as one of the main challenges to be addressed for future generations of BIGDML. Nevertheless, the BIGDML framework is envisioned as a base model to further develop towards constructing more general and efficient force fields for materials modelling. Two possible avenues to address transferability are: (1) to use a construction approach where smaller models are combined to approximate larger supercells and (2) Localize the global descriptor to simplify interactions. In the first approach, models trained on small supercells, could be combined to span larger supercells in an approximate way by

means of a convolution approach: $\hat{f}_{large} \approx \rho_{large} * \hat{f}_{small}$, where $\rho_{large}$ is the atomic distribution in the larger supercell, this being one approach towards larger and transferable composite models. In the second option, the mathematical form of the BIGDML predictor can be reformulated by, for example, reducing the many-body complexity of the $\mathcal{D}^{PBC}$ descriptor to keep interactions only up to a certain body order. This approach has been proven to give good results in molecules[102,103]. Despite current limitations of BIGDML, having access to a MLFF that can robustly represent global interactions in extended materials is a substantial achievement, as shown via extensive simulations in Section. In addition, we should stress that BIGDML has a superior learning capacity compared to local MLFFs, since it can reach generalization accuracies of up to two orders of magnitude better than localized MLFFs (see Fig. 3).

Another crucial aspect of MLFFs is their data efficiency and ability to correctly capture all relevant symmetries for a given system. Symmetries play a crucial role when studying nuclear displacements (phonons, thermal conductivities, etc). BIGDML addresses both of these challenges at the same time. The symmetries are obtained from the periodic cell and the reference geometries in a data-driven way. Symmetries are known to effectively reduce the complexity of the learning problem (cf. refs. [46,104]), as we have shown by introducing energy conservation (i.e. time homogeneity)[42] and molecular point groups for finite molecular systems[23]. Periodic systems have even more symmetries than molecules, making the force field reconstruction effectively a lower-dimensional task. While this qualitative outcome could have been expected prior to the formulation of BIGDML, the enormous practical advantage of incorporating crystalline symmetries is remarkable. Even a few dozen samples (atomic forces for a few unit cell geometries) already yield BIGDML models that can be used in practical applications of molecular dynamics.

We would like to remark further that while BIGDML is a kernel-based approach (see e.g. refs. [105–108]), able to include symmetries and prior physical information, it will be an interesting and important challenge to transfer the learning machinery established here also to deep learning approaches (such as convolution neural networks, graph neural networks or even generative adversarial models) ideally by incorporating symmetries, prior physical knowledge and equivariance constructions into their architecture (see refs. [24,34,109,110] for some first steps in this direction).

In this work, we have focused on materials with fixed supercell size and shape, nevertheless, a large number of physical phenomena in materials involve phase transitions and symmetry breaking. These systems represent a challenge to be addressed and requires developing further the ideas introduced in the BIGDML model. In this regard, the mathematical structure of the BIGDML framework has the foundations to allow the study of systems with flexible supercell (i.e. changing supercell volume and/or lattice vectors). This is because the defined metric (Euclidean distance between two structures) by the global representation does not depend on the particular selection of the lattice vectors. Meaning that, if we have two structure configurations $X_1$ and $X_2$ with lattice vectors $a_1$ and $a_2$, respectively, their distance in descriptor space $\| \mathcal{D}^{(PBC)}_{lattice_1}(X_1) - \mathcal{D}^{(PBC)}_{lattice_2}(X_2)\|$ is well-defined. Hence, exploiting such invariance in the metric could allow the description of materials with fluctuating lattice vectors, as it would be the case, for example, in simulations of materials described by the $NpT$ ensemble.

Another challenge to be addressed is the need to describe even larger systems. In Section -D, we have already shown that it is possible to accurately describe highly fluxional structures with supercells containing 160 atoms and trained on up to 1000 structures. Given this evidence, the only limitation when moving to materials containing hundreds of atoms per supercell is memory requirements, a problem that is solved by using numerical solvers as it is done for training neural networks (see e.g. ref. [111]). Nevertheless, the BIGDML framework requires multiple extensions to scale up to much larger systems (with thousands of atoms per unit cell) or systems with a larger number of symmetries.

With the advent of new advanced materials such as high performance perovskite solar cells, topological insulators and van der Waals materials, it is crucial to construct reliable MLFFs capable of dynamical simulations at the highest level of accuracy given by electronic-structure theories while maintaining relatively low computational cost. While local MLFFs and BIGDML are complementary approaches, we would like to emphasize that global representations and symmetries could also be readily incorporated in other MLFF models. The challenge of developing accurate, efficient, scalable, and transferable MLFFs valid for molecules, materials, and interfaces thereof suggests the need for many further developments aiming towards universally applicable MLFF models.

## Methods

**Data generation and DFT calculations**. Given the different types of calculations and materials in this work, we present the details of the data generation, model training and simulations organized per system. All the databases were generated using molecular dynamics simulations using the NVT thermostat.

*Graphene*. Here we used a $5 \times 5$ supercell at the DFT level of theory at the generalized gradient approximation (GGA) level of theory with the Perdew-Burke-Ernzerhof (PBE)[91] exchange-correlation functional, We performed the calculation in the Quantum Espresso[112,113] software suite, using plane-waves with ultrasoft pseudopotentials and scalar-relativistic corrections. We used an energy cutoff of 40 Ry. A uniform $3 \times 3 \times 1$ Monkhorst-Pack grid of $k$-points was used to integrate over the Brillouin zone. The ab initio MD (AIMD) used to generate the database was ran at 500 K during 10,000 time steps using an integration step of 0.5 fs. The results displayed in Fig. 7 were performed using PIMD simulations with 32 beads, and we ran the simulation for 300 ps using an integration step of 0.5 fs.

*Pd$_1$/MgO*. In this case, we used a $2 \times 2$ supercell with 3 atomic layers to model the MgO (100) surface. The calculations were performed in Quantum Espresso, using an energy cutoff of 50 Ry and integrating over the Brillouin zone at the Γ-point only. For this system, we ran an AIMD at 500 K with an integration step of 1.0 fs during 10,000 integration steps to generate the material's database.

*Benzene/graphene*. For this particular example, we have used the same graphene supercell mentioned above and placed a benzene molecule on top. In order to include the correct non-covalent interactions between the benzene molecule and the graphene layer, we have used an all-electrons DFT/PBE level of theory with the many body dispersion (MBD)[69,70] treatment of the van der Waals interaction using the FHI-aims[114] code. The AIMD simulation for the system's database constructions was performed at 500 K using an integration step of 1.0 fs during 15,000 steps. The results displayed in Fig. 8 were performed using PIMD simulations using 1, 8, 16 and 32 beads (in order to guarantee that we have achieved converged NQE) and we ran the simulation for 200 ps using an integration step of 0.5 fs.

*Bulk metals*. In this case, we were interested in a variety of materials and their different interactions. Then, we have considered Pd[FCC] and Na[BCC] described at the DFT/PBE level of theory using the Quantum Espresso software. The databases were created by running AIMD simulations at 500 K and 1000 K for Pd, and 300 K for Na using a time steps of 1.0 fs for all the simulations. Monkhorst-Pack grids of $3 \times 3 \times 3$ $k$-points were used to integrate over the Brillouin zone for all materials. All calculations for the bulk metals were spin-polarized.

*H in Pd[FCC]*. In this case we used a supercell of $3 \times 3 \times 3$ with 32 Pd atoms and a single hydrogen atom described by DFT/PBE level of theory using the Quantum Espresso software. The database was generated by running AIMD at 1000 K. We used time steps of 1.0 fs and a total dynamics of 6 ps. Monkhorst-Pack grids of $3 \times 3 \times 3$ $k$-points were used to integrate over the Brillouin zone for all materials. The results shown in Fig. 9 we obtained by running classical MD and PIMD simulations using an interface of the BIGDML FF with the i-PI simulation package[115]. We ran the simulations at various temperatures from 300 K to 1000 K. In each case, we employed a time step of 2.0 fs during 2,000,000 steps, for a total simulation time of 4 ns. For the PIMD simulations we used a different number of beads for each temperature: 32 for 100, 300, and 600 K; 24 for 400 K; 2 for 700 K; and 4 for 800 K. Using this data we were able to compute the H diffusivity as a function of the temperature.

**The sGDML framework**. A data efficient reconstruction of accurate FFs with ML hinges on including the right inductive biases in the model to compensate for finite reference dataset sizes. The Symmetric Gradient-Domain Machine Learning (sGDML) framework achieves this through constraints derived from exact physical laws[23,42,43]. In additional to the basic roto-translational invariance of energy, sGDML implements energy conservation, a fundamental property of closed classical and quantum mechanical systems. The key idea behind sGDML is to define a Gaussian Process (GP) using a kernel $\mathbf{k}(\mathbf{x}, \mathbf{x}') = \nabla_{\mathbf{x}} k_E(\mathbf{x}, \mathbf{x}')\nabla_{\mathbf{x}'}^{\top}$ that models any force field $\mathbf{f_F}$ as a transformation of some unknown PES $f_E$ such that,

$$\mathbf{f_F} = -\nabla f_E \sim \mathcal{GP}\left[-\nabla \mu_E(\mathbf{x}), \nabla_{\mathbf{x}} k_E(\mathbf{x}, \mathbf{x}')\nabla_{\mathbf{x}'}^{\top}\right]. \quad (2)$$

Here, $\mu_E : \mathbb{R}^d \to \mathbb{R}$ and $k_E : \mathbb{R}^d \times \mathbb{R}^d \to \mathbb{R}$ are the prior mean and prior covariance (kernel) functions of the latent energy GP-predictor, respectively. Regarding the functional form of the kernel, the sGDML framework uses the Matérn covariance function with restricted differentiability to second order: $k_E(d) = \sigma^2(1 + d + \frac{1}{3}d^2)e^{-d}$, $d = \frac{\sqrt{5}|\mathbf{x} - \mathbf{x}'|}{\rho}$, where $\sigma$ and $\rho$ are the normalization and scale parameters, respectively.

Each molecular geometry $\mathbf{R}$ represented in descriptor space by $\mathbf{x} = \mathcal{D}^{(PBC)}(\mathbf{R})$ is encoded using the proposed descriptor (see Eq. (1)). A sGDML FF for a particular system is then obtained by solving a linear system for $\overrightarrow{\alpha}$,

$$(\mathbf{K} + \lambda \mathbb{1})\overrightarrow{\alpha} = -\mathbf{F}, \quad (3)$$

where the set of $\overrightarrow{\alpha}$s are the trainable parameters, $\mathbf{K}_{ij} = \mathbf{k}\left(\mathbf{x}_i, \mathbf{x}_j\right)$ and $\mathbf{F} = -\nabla V_{BO}$ are the gradients of the PES as specified by the corresponding reference calculations. By construction of the kernel matrix, the resulting model is guaranteed to be integrable, such that the corresponding PES is recovered by

$$\int \mathbf{f_F} \, d\mathbf{R} = -f_E + c \quad (4)$$

up to an integration constant $c$.

The sGDML model additionally incorporates all relevant rigid space group symmetries, as well as dynamic non-rigid symmetries of the system at hand. This is achieved via marginalization of the kernel over the permutation set $\pi \in S$:

$$\mathbf{f_F}(\mathbf{x}) = \sum_{i=1}^{M} \overrightarrow{\alpha}_i \sum_{\pi \in S} \mathbf{k}\left(\mathbf{x}, \mathbf{P}_{\pi}\mathbf{x}_i\right). \quad (5)$$

Here, M is the number of training datapoints and $P_{\pi}$ is the permutation operator in descriptor space. In the original model, these symmetries are automatically recovered as atom-permutations via multi-partite matching of all geometries in the training dataset[23]. BIGDML supplements this set by adding permutational symmetries that are unique to periodic systems and were previously not considered.

Now, some aspects on training and deploying sGDML FFs. Solving the linear system Eq. (3) is the computationally most strenuous aspect of the training procedure, as it incurs a cost of $\mathcal{O}((3NM)^3)$. Moreover, BIGDML is trained in closed-form (via matrix decomposition), which requires storing the kernel matrix at $\mathcal{O}((3NM)^2)$ memory cost. The inclusion of symmetries only incurs extra linear cost during kernel matrix construction in this scenario, while the training cost remains the same. These economics are ideal for the application to periodic systems, where we can impose a strong inductive prior through the inclusion of large symmetry sets, which allows the number of training points $M$ to remain small. Now, in cases where memory limitations appear, the model can be trained by a numerical solver as in the case of neural networks. This approach allows training much larger models and bigger systems.

**Coulomb matrix PBC implementation**. The periodic boundary conditions were implemented using the minimum image convention. Under this convention, we take the distance between two atoms to be the shortest distance between their periodic images. We start by expressing the distance vectors $\mathbf{d}_{ij} = \mathbf{r}_i - \mathbf{r}_j$ in the basis of the simulation supercell lattice vectors as

$$\mathbf{d}_{ij} = \mathbf{A}\mathbf{c}_{ij}, \quad (6)$$

where A is a $3 \times 3$ matrix which contains the lattice (supercell) vectors as columns, and $c_{ij}$ are the distance vectors in the new basis. We then confine the original distance vectors to the simulation cell,

$$\mathbf{d}_{ij}^{(PBC)} = \mathbf{d}_{ij} - \mathbf{A}\text{nint}(\mathbf{c}_{ij}), \quad (7)$$

where nint(x) is the nearest integer function. By replacing the ordinary distance vectors $d_{ij}$ with $\mathbf{d}_{ij}^{(PBC)}$ in the Coulomb matrix descriptor, it becomes

$$\mathcal{D}_{ij}^{(PBC)} = \begin{cases} \frac{1}{|\mathbf{d}_{ij}^{(PBC)}|} & \text{if } i \neq j \\ 0 & \text{if } i = j \end{cases} \quad (8)$$

In practice, only the $d^{(PBC)}$ upper triangular matrix is used.

**Software: Interface with i-PI**. For this work, a highly optimised interface of BIGDML has been implemented in the i-PI molecular dynamics package[115]. The main features of this implementation are: (1) it allows the use of periodic boundary conditions and stress tensor calculation, (2) parallel querying of all beads at once in PIMD simulations and (3) it uses the highly optimized sGDML GPU implementation in PyTorch to parallelise beads calculations, dramatically increasing the simulation efficiency.

**Software: interface with phonopy for phonons**. An ASE calculator is already provided by the sGDML package, this allows to use all its simulation options. In particular, the phonon analysis for materials is easily computed in this package using Phonopy[116]. An example of the scripts used to compute the phonons in this paper is provided in the Supplementary Software.

**Training GAP/SOAP models**. The GAP models for graphene were trained employing the QUIP software package available at http://www.libatoms.org. All potentials were constructed using the same training datasets prepared to train the BIGDML models in this work. A combination of a two-body (2b), a three-body (3b) and a many-body (SOAP) descriptor were used in the construction of each GAP model. The parameters for the 2b, 3b and SOAP descriptors were the same optimized values used in the work of Rowe et al. to fit their GAP model for graphene[36].

# Data availability

All datasets used in this work are available at http://www.sgdml.org or http://quantum-machine.org/datasets/. Additional data related to this paper may be requested from the authors.

# Code availability

The full documentation of the sGDML software can be found at, http://quantum-machine.org/gdml/doc/ and the code can be downloaded from https://github.com/stefanch/sGDML.

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

## Acknowledgements

This paper is dedicated to the memory of Astro. We thank Oz Yosef Mendelsohn and Leeor Kronik for providing access to the perovskite dataset. AT was supported by the Luxembourg National Research Fund (DTU PRIDE MASSENA) and by the European Research Council (ERC-CoG BeStMo). KRM was partly supported by the Institute of Information & Communications Technology Planning & Evaluation (IITP) grants funded by the Korea government(MSIT) (No. 2019-0-00079, Artificial Intelligence Graduate School Program, Korea University and No. 2022-0-00984, Development of Artificial Intelligence Technology for Personalized Plug-and-Play Explanation and Verification of Explanation), and by the German Ministry for Education and Research (BMBF) under Grants 01IS14013A-E, 01GQ1115, 01GQ0850, 01IS18025A and 01IS18037. LOPB thank financial support from DGAPA-UNAM (PAPIIT) under Projects IA102218 and IN116020, as well as cpu-time at Supercómputo UNAM (Miztli) through a DGTIC-UNAM grant LANCAD-UNAM-DGTIC-307. LOPB and LEGG are also grateful to CONACYT-Mexico for support through Project 285218 and a doctoral scholarship 493775, respectively. HES works at the BASLEARN - TU Berlin/BASF Joint Lab for Machine Learning, co-financed by TU Berlin and BASF SE. HES thanks the support from DGTIC-UNAM under Project LANCAD-UNAM-DGTIC-419. Correspondence should be addressed to HES, KRM and AT.

## Author contributions

H.E.S. and A.T. conceived the research and designed the analyses. H.E.S. and L.E.G.G. performed quantum chemical calculations and the molecular dynamics simulations. S.C. conceived and constructed the GDML framework. S.C., L.E.G.G., H.E.S., A.T., and K.-R.M. developed the machine learning methodology. H.E.S. and L.E.G.G. created the figures with help from other authors. H.E.S., K.-R.M. and A.T. wrote the paper. All the authors discussed results and commented on the manuscript.

## Funding

## Competing interests

The authors declare no competing interests.
