## [Peer Review File · Nature Communications]

REVIEWER COMMENTS

Reviewer #1 (Remarks to the Author):

In this work, the authors present a machine-learning-based methodology for constructing the potential energy surface of materials. Compared with existing approaches, despite some new components like the Bravais inspired construction, I don't see much real applicability of this method to complicated situations, neither broader impacts on the field. As such, in my view, this work does not meet the publication standard of Nature Communications and should be rejected.

In detail, methodology-wise, as the authors claim, "BIGDML model is system-specific and, hence, not transferable between different systems or even between different supercell sizes for the same system. Despite this slight drawback, ..." I would say that this is not a slight drawback. This "slight drawback" will prevent the methodology from most useful applications. For example, in the case of hydrogen interstitial in bulk palladium, at different temperatures, the pressure cannot even be controlled and the equilibrium volume cannot be changed. This presents a big problem in the predicted properties. In addition, the authors compare with GAP, but I don't think GAP is a good model to be compared with, since it requires a trial-and-error process, and the failure of GAP shouldn't imply the failure of a local descriptor compared with a global one. Is this a fitting error or prediction error? What would be the performance of other more automatic methods, say, SchNet that has been developed by part of the authors?

Regarding the applications, they give me an impression that they are designed for this methodology, but do not represent practically challenging cases. The authors have to really tell the reader what should be important applications that the method can be applied to. Besides, I don't think the H diffusivity, a dynamical property, can be properly extracted from a PIMD simulation. The nuclear quantum effects on dynamical properties should be treated more carefully with, e.g., RPMD, or LSC-IVR-related approaches.

Reviewer #2 (Remarks to the Author):

The paper "BIGDML: Towards Exact Machine Learning Force Fields for Materials" by Saucedo et al. introduces a method to construct machine learning force fields using ab initio data. This is a very important and active field of research nowadays. At variance with mainstream methods that make use of local descriptors, this work employs global descriptors and explicit consideration of the

symmetries of crystalline solids. The authors find that this approach leads to errors in the energy that are up to an order magnitude smaller than in other methods. They also present many examples to illustrate the usefulness of their approach.

The technical part of this work is very good. The paper is well-structured and well-written. The figures are both informative and aesthetically pleasing. It seems to me that enough details are provided to reproduce the results.

This work presents an extension of the approach described in the papers Chmiela et al., Nat. Commun. 9, 3887 (2018) and Chmiela et al., Comput. Phys. Commun. 240, 38 (2019). Previous work dealt with molecules and this manuscript considers the construction of force fields for crystalline solids. One of the previous articles by the authors was already published in Nature Communications.

In my opinion the claims of this paper might be exaggerated to some extent and for this reason the manuscript needs revision. On the one hand, the use of the word "exact" to describe an improvement in performance is misleading. On the other hand, the manuscript highlights the advantages of this method but sweeps under the rug the many disadvantages. There is nothing wrong with a method not being perfect, but the publication of a biased account of the facts in a high profile journal would be troublesome. These are the disadvantages that I think the manuscript fails to describe in a balanced fashion:

- 1) The method presented here works only for very small systems and only for one system size. I find this to be a tremendous limitation. Most methods work for any system size and leverage the ability to train on small systems and simulate very large systems. Methods that use local descriptors also have linear scaling with the number of nuclei while the method presented here would have quadratic scaling.

- 2) The use of symmetries is highlighted in this work. However, I find that the relevance of most of these symmetries stems from the use of a global descriptor. Methods that employ local descriptors automatically include invariance with respect to translations, and are typically permutation invariant.

- 3) I think that the most relevant symmetries included in this work are those of the point group of the crystalline structure. However, these particular symmetries can only be used if a single crystalline phase is considered. Therefore the method works only in this particular scenario. Many important problems also involve the simulation of the liquid phase (crystallization or solid-liquid interfaces) or several polymorphs (polymorph interconversion). I believe this method does not work in these important cases and therefore it has somewhat limited applicability. Current machine learning force fields might be less accurate but are general - they can typically reproduce the properties of any phase if trained properly.

4) The improvement in the energy and forces accuracies seems to be valid only for force fields that describe a single crystal structure.

In spite of these limitations, the manuscript contains relevant results that are worth publishing and could be of significance to the field. It presents a different avenue for constructing machine learning force fields that could be important even if restricted to the particular case of crystalline solids. I recommend accepting this manuscript in Nature Communications after extensive revision such that the limitations above can be clearly and easily understood by a reader. Please, do not bury this discussion in the last paragraphs of the paper.

Other comments:

1) I think the manuscript is missing an explanation of some important details. The functional form used to learn the forces is only given in Figure 1c. It would be good to include it in the text and give a thorough explanation. Otherwise the reader is forced to read references 22 and 41.

2) Another important question is whether the improvement of the accuracy of the energy and forces achieved by this method is a consequence of the use of a long range descriptor or the inclusion of symmetries. Please try to answer this question.

3) It is not clear to me how the curves for GAP/SOAP shown in Figure 3 were obtained. Did you obtain the data from a paper? Did you train a GAP/SOAP model using the training set you prepared for BIGDML? Please clarify.

4) The word exact is used both in the title and in the discussion. Please refrain from using that term. I suggest changing it to "highly accurate" or something similar.

Reviewer #3 (Remarks to the Author):

This paper introduces a statistical learning method ("BIGDML") for the energy and atomic forces of bulk systems with symmetries, followed by a number of demonstrations on systems including bulk and 2D materials, surface absorption, and interstitial hydrogen in a metal. The paper is well-written and the ideas are clear. My main concern is about the applicability and the merit of the method. In my opinion, it is not clear whether their method is advantageous over the state-of-the-art even in a subset of situations. My reasoning is given below.

The method has two main components in addition to the features of sGDML that was introduced by the same group of authors: the first is using a minimal image convention when computing the Coulomb matrices, which then serves as the descriptors of a system. The second is exploiting space group symmetry.

These two components both introduce shortcomings, however. First and foremost, because a global descriptor is used, and the size of the descriptor scales quadratically with the number of atoms, the system size is severely restricted, to about <100 atoms. Such small sizes are clearly too small for most applications when modeling moderately complex systems. Second, the space group symmetry can only be exploited when the system contains such symmetries, which means that only a small fraction of systems will benefit from the methodology. Moreover, whenever interesting phenomena happen, such as a breaking of a bond in graphene, or a formation of any types of crystalline defects in bulk materials, or a surface reconstruction, the symmetries no longer apply.

On the other hand, the advancement that is demonstrated in the paper on a few example systems can be easily achieved using state-of-the-art methods. None of the applications in the paper couldn't have been done using pure DFT or standard ML potentials. Training seems to be more efficient, when comparing with the GAP models that they have fitted, but at the expense of having to use the same (small) system size when training the model and performing the simulations. It is also not clear how BIGDML will perform when there are crystalline defects that break the space group symmetry. It is of course okay to introduce a new method with strengths and weaknesses, but in this case, how and whether the new method is advantageous over the standard ones is not clearly demonstrated from the examples.

The authors offer some reasons why their method is better, which are not very convincing. In the introduction, the authors state that their method is better because it is a global model without the locality assumption. However, since their system sizes are small, one can just use a BPNN/GAP/DeepMD/... with an atomic cutoff radius larger than the supercell length, or a small cutoff combined with a few message passing operations in NequIP/SchNet to achieve a global model. Another reason is that "additional advantage of a global representation is that cross-correlations between forces on different atomic species are dealt with rigorously, at variance with existing atomic representations." But this is not clear to me and should be expanded. They say that "Our choice of CM with PBC enforced using MIC is the simplest and most efficient choice", but it is not clear why their choice is better than the Sine matrix or CM combined with an Ewald-sum.

I also have some doubts about the novelty of the method itself. The minimal image convention is used through and through in any simulation studies involving periodic boundary conditions. The exploitation of the discrete symmetry is equivalent to data augmentation, which has been widely used.

Minor points:

It is probably much clearer to draw the learning curves using the log scale on the x-axis.

More information on how the GAP/SOAP models were fitted should be given. It is not clear whether reasonable GAP parameters were chosen for these fits.

The acronym "BIGDML" is misleading as it hints at large-scale ML models but in fact both the training points and the typical system sizes are small.

REVIEWER COMMENTS

Reviewer #1 (Remarks to the Author):

In this work, the authors present a machine-learning-based methodology for constructing the potential energy surface of materials. Compared with existing approaches, despite some new components like the Bravais inspired construction, I don't see much real applicability of this method to complicated situations, neither broader impacts on the field. As such, in my view, this work does not meet the publication standard of Nature Communications and should be rejected.

In detail, methodology-wise, as the authors claim, "BIGDML model is system-specific and, hence, not transferable between different systems or even between different supercell sizes for the same system. Despite this slight drawback, ..." I would say that this is not a slight drawback. This "slight drawback" will prevent the methodology from most useful applications. For example, in the case of hydrogen interstitial in bulk palladium, at different temperatures, the pressure cannot even be controlled and the equilibrium volume cannot be changed. This presents a big problem in the predicted properties.

Response 1-1): *First, we would like to thank the reviewer for carefully reading our manuscript and for the suggestions to improve the article.*

We politely disagree with the view of the referee that our manuscript does not meet the standard of Nature Communications. We remark that the developed BIGDML framework is the first approach for constructing machine-learned force fields (MLFF) for extended materials that does not rely on the ad hoc locality approximation (or sum over atomic energies). Furthermore, we demonstrate the applicability of BIGDML to low-dimensional and bulk materials, defects in solids, and atoms and molecules interacting with surfaces. This broad set of presented applications and the obtained combination of accuracy and efficiency of the BIGDML approach goes much beyond the current state of the art in the construction of MLFF's for extended materials.

As the reviewer mentioned, there are particular study cases where performing simulations of the extended system requires going beyond the NVT/NVE ensemble. Nevertheless, there are a plethora of applications of ab initio MD simulations for materials where the NVT/NVE ensemble is used. Some immediate examples are finite temperature studies of fluxional solids (e.g. organic-inorganic perovskites), molecule@surface systems, surface catalytic reactions, interfaces, etc. All those systems are suitably described by the BIGDML model.

Furthermore, there is no inherent theoretical limitation within our model that restricts the use of flexible supercells (i.e. changing volume supercell and dynamic lattice vectors). By construction, the kernel used by the BIGDML model is capable of using different internal geometries and lattice vectors for each training point. This result is analogous to the fluxional behaviour of molecular systems and how the sGDML model deals with them.

In order to make this point clearer to the reader, we have added a discussion (section III) on how the BIGDML approach can also be used for systems beyond the NVT/NVE ensembles.

In addition, the authors compare with GAP, but I don't think GAP is a good model to be compared with, since it requires a trial-and-error process, and the failure of GAP shouldn't imply the failure of a local descriptor compared with a global one. Is this a fitting error or prediction error? What would be the performance of other more automatic methods, say, SchNet that has been developed by part of the authors?

Response 1-2): Here, the referee makes an interesting point about GAP usability; it is a model that requires some hand-tuning, meaning it depends also on human intuition and intervention. Contrasting this approach, our BIGDML model is fully automated and can be used out-of-the-box, achieving already the final accuracy for the given system without fine-tuning. On the technical part of Fig. 3, we show test errors (i.e. prediction errors).

In our article we are interested in comparing to what is considered in the literature as one of the state-of-the-art models for materials, as well as to allow a reasonable comparison between two kernel-based models.

Action taken:

We have extended this explanation in Methods section in a new subsection "Training GAP/SOAP models".

In the case of SchNet, this is not a methodology that has been used to perform MD simulations of materials up to now. However, we can compare to molecular datasets such as the MD17, where a global descriptor in sGDML is clearly more advantageous compared to the SchNet local descriptor [NIPS2017 (<https://arxiv.org/pdf/1706.08566>)].

Regarding the applications, they give me an impression that they are designed for this methodology, but do not represent practically challenging cases. The authors have to really tell the reader what should be important applications that the method can be applied to. Besides, I don't think the H diffusivity, a dynamical property, can be properly extracted from a PIMD simulation. The nuclear quantum effects on dynamical properties should be treated more carefully with, e.g., RPMD, or LSC-IVR-related approaches.

Response 1-3): The examples selected in the article are on one hand benchmark systems (such as graphene and cubic lattice solids) and other more challenging systems that present known problematic features when describing them by ML force fields (benzene/graphene, Pt at MgO and H in Pd). On one side, multielement systems are known to challenge the performance of GAP and other kernel based methods, while fluxional systems are problematic for even the state-of-the-art NNs given that atomic representations have to encode the embeddings of contrasting environments as well as to capture very subtle structural fluctuations such in the case of non-covalent interactions. Many of these systems have not yet been convincingly addressed with machine-learned force fields.

Regarding the defect calculations, the statistical dynamical effects are correctly described by PIMD since we focus on the diffusion coefficient, which only requires sampling the PES and can be accurately computed when considering the centroid of the ring polymer.

We note in passing that Fig. 8 and the associated discussion have been revised to reflect a more accurate methodology to compute the hydrogen diffusivity using the mean-square displacement analysis. Additionally, we used this methodology to reproduce the experimental results published by Volk(1971), Powell(1991), and Heuser(2014), providing further support to the accuracy of our methodology.

Action taken:

We have extended the discussion regarding H diffusivity, section II-F-2 and updated the figure with results comparing our calculated results to experimental values.

Reviewer #2 (Remarks to the Author):

The paper "BIGDML: Towards Exact Machine Learning Force Fields for Materials" by Saucedo et al. introduces a method to construct machine learning force fields using ab initio data. This is a very important and active field of

research nowadays. At variance with mainstream methods that make use of local descriptors, this work employs global descriptors and explicit consideration of the symmetries of crystalline solids. The authors find that this approach leads to errors in the energy that are up to an order magnitude smaller than in other methods. They also present many examples to illustrate the usefulness of their approach.

The technical part of this work is very good. The paper is well-structured and well-written. The figures are both informative and aesthetically pleasing. It seems to me that enough details are provided to reproduce the results.

This work presents an extension of the approach described in the papers Chmiela et al., Nat. Commun. 9, 3887 (2018) and Chmiela et al., Comput. Phys. Commun. 240, 38 (2019). Previous work dealt with molecules and this manuscript considers the construction of force fields for crystalline solids. One of the previous articles by the authors was already published in Nature Communications.

In my opinion the claims of this paper might be exaggerated to some extent and for this reason the manuscript needs revision. On the one hand, the use of the word "exact" to describe an improvement in performance is misleading.

Response 2-1): *We agree with the referee that our approach cannot be considered "exact" at this moment for materials, and this explains our usage of a qualifier "Towards" in the title of our paper. Our BIGDML approach would become exact only when "gold standard" calculations [e.g. at CCSD(T) or QMC level] become available for extended systems. For example, for molecules such data is available and essentially exact MD simulations for small molecules are now within reach [see our previous publication <https://www.nature.com/articles/s41467-018-06169-2>]. Since the only remaining stepping stone to "Exact MD" for materials is the availability of high-quality reference data, we would consider the title of our manuscript a reasonable way to communicate our findings.*

On the other hand, the manuscript highlights the advantages of this method but sweeps under the rug the many disadvantages. There is nothing wrong with a method not being perfect, but the publication of a biased account of the facts in a high profile journal would be troublesome. These are the disadvantages that I think the manuscript fails to describe in a balanced fashion:

1) The method presented here works only for very small systems and only for one system size. I find this to be a tremendous limitation. Most methods work for any system size and leverage the ability to train on small systems and simulate very large systems. Methods that use local descriptors also have linear scaling with the number of nuclei while the method presented here would have quadratic scaling.

Response 2-2): *We thank the reviewer for highlighting the size of the system and the computational scaling of the model.*

We first remark that the developed BIGDML framework is the first approach for constructing machine-learned force fields (MLFF) for extended materials that does not rely on the uncontrolled locality approximation (or sum over atomic energies). The solution of the Schrödinger equation can not be in general represented as a sum over atomic energies and, so far, all models for materials known to us employed the locality approximation. We demonstrate the applicability of BIGDML to low-dimensional and bulk materials, defects in solids, and atoms and molecules interacting with surfaces. This broad set of presented applications and the obtained combination of accuracy and efficiency of the BIGDML approach goes much beyond the current state of the art in the construction of MLFF's for extended materials.

Regarding system sizes for BIGDML models: In principle, there are no limitations for the BIGDML framework regarding the size of the systems it can describe. The computational requirements during training (i.e. specifically the RAM memory) can indeed become high when using an analytical solver for larger systems, but this problem can be addressed by using a numerical solver as it is done for training neural networks. Specifically in our case, this is work in progress and will be released in a separate publication where we train

the model on systems with hundreds of atoms. Hence, this removes the limitation that the model can be trained only on small systems.

Action taken:

We have added a paragraph in the introduction mentioning this issue.

“It is important to mention that describing materials having several hundreds of atoms or have an exceedingly large amount of symmetries still remain challenging tasks for the model. Nevertheless, these technical issues could be addressed by utilising multiscale and composite approaches as well as numerical solvers (see section III for an extended discussion).”

Added a more extended discussion in section III of the paper.

“... Additionally, the construction of models for materials containing several hundreds of atoms per supercell could be done by using numerical solvers to overcome memory limitations as it is done for training neural networks (see e.g. LeCun(2012)).”

Regarding supercell sizes: In many-body problems, such as electronic structure, it is well known that the total energy of the system cannot be partitioned in atomic contributions given the delocalised nature of the wavefunction. Thereby, the BIGDML model uses a global representation of the system as an inductive bias to preserve the non-trivial many-body nature of quantum interactions. The BIGDML approach supports the fact that electronic interactions cannot be localized in general, and offers a unique solution to learning the different types of interactions in extended materials. There is indeed a limitation on scalability, and it is the topic for future work.

*On a more general note, as mentioned in **response 2-1**, the BIGDML model is meant to be a base model for further developments towards constructing more general and efficient models. For example, models trained on small supercells can be combined to span larger supercells in an approximate way by means of a convolution approach. This will directly address the two challenges mentioned by the reviewer.*

Regarding the scaling with the number of atoms N , the BIGDML model is trained in closed-form (by solving the associated linear system directly, via matrix decomposition), which scales as $O((3NM)^2)$ (M is the number of training points). Nevertheless, the small prefactor and the simplicity of evaluating the model (Matrix-vector multiplication) make the model highly parallelizable, which is fully exploited in our GPU implementation. In this way, even for systems with more than hundred atoms, a model evaluation cost still remains in the order of tens of milliseconds.

To address the reviewer’s question, we have extended the discussion (section III) regarding the system sizes that the model can describe, as well as its limitations in the perspective part of section III. The scaling of the BIGDML’s performance with the number of atoms has been added in the extended description of the sGDML model (in the Methods section).

2) The use of symmetries is highlighted in this work. However, I find that the relevance of most of these symmetries stems from the use of a global descriptor. Methods that employ local descriptors automatically include invariance with respect to translations, and are typically permutation invariant.

Response 2-3): *Indeed, given the nature of a global descriptor, the use of the relevant symmetries of the system is required to generate a symmetrized kernel. The use of a global descriptor is required to preserve the non-local origin of interatomic quantum interactions, which then avoids a non-unique partitioning of the total energy of the system. Furthermore, as stated in the article, the explicit use of only the necessary symmetric operations avoids many of the limitations faced by local descriptors originated while in pursuit of permutational invariance. In Figure 1 of the article, we display how the different elements in the BIGDML model contribute to the accuracy of the model and how each sub-group of symmetries reshape the potential energy surface.*

3) I think that the most relevant symmetries included in this work are those of the point group of the crystalline structure. However, these particular symmetries can only be used if a single crystalline phase is considered. Therefore the method works only in this particular scenario. Many important problems also involve the simulation of the liquid phase (crystallization or solid-liquid interfaces) or several polymorphs (polymorph interconversion). I believe this method does not work in these important cases and therefore it has somewhat limited applicability. Current machine learning force fields might be less accurate but are general - they can typically reproduce the properties of any phase if trained properly.

Response 2-4): *The individual contributions of the translational and point group symmetries to the accuracy of the model depend on the specific case at hand. For example, in the case of Pt/MgO (as displayed in Fig. 1-vi) the point group of the system contributes more, but in cases such as cubic crystals, translational symmetries are the leading contribution for the accuracy of the predictions.*

Regarding phase transitions, even though those cases are beyond the scope of our current paper, there is no fundamental limitation in the BIGDML model to describe them. Two different crystalline phases in descriptor space are simply two domain regions (each having its own set of symmetries) separated by one or more transition paths. This case scenario is not different from the molecular case, where two local minima of the molecule have two different symmetry groups and are perfectly represented by a global descriptor, as we have extensively demonstrated in previous publications. A simple example is the ethanol molecule, which has three minima, gauche(+/-) and trans, with point groups C1 and Cs, respectively.

Action taken:

We have added a paragraph in the Discussion section where we analyse this in detail.

4) The improvement in the energy and forces accuracies seems to be valid only for force fields that describe a single crystal structure.

Response 2-5): *There is no fundamental limitation for our model to describe materials with multiple crystalline structures, as mentioned in the previous response. In descriptor space, multiple crystalline phases are just multiple clusters of data connected by transition paths. This scenario is perfectly described by our model.*

Action taken:

We have added a paragraph in the Discussion section where we analyse this in detail.

In spite of these limitations, the manuscript contains relevant results that are worth publishing and could be of significance to the field. It presents a different avenue for constructing machine learning force fields that could be important even if restricted to the particular case of crystalline solids. I recommend accepting this manuscript in Nature Communications after extensive revision such that the limitations above can be clearly and easily understood by a reader. Please, do not bury this discussion in the last paragraphs of the paper.

Other comments:

1) I think the manuscript is missing an explanation of some important details. The functional form used to learn the forces is only given in Figure 1c. It would be good to include it in the text and give a thorough explanation. Otherwise the reader is forced to read references 22 and 41.

Response 2-6): *Part of this explanation was already included in section IV-2 of the article. We have extended such a description to provide a more self-contained structure to the article (Methods section-2).*

2) Another important question is whether the improvement of the accuracy of the energy and forces achieved by this method is a consequence of the use of a long range descriptor or the inclusion of symmetries. Please try to answer this question.

Response 2-7): *This is related to Response 2-4. In general, the achieved accuracy is a combination of complementary contributions. The Coulomb matrix analytical (long range) form provides a global description of the system as a whole, the Matern kernel provides a basis function that correctly describes the tails of the contributions of each training point, and the full crystal symmetry group reduces the dimensionality of the predictor function. This can be seen in Fig. 1-D-vi, where the precision of the models has consistent increments as each element is added.*

Action taken:

We have added several remarks in this regard at the end of section II-D.

3) It is not clear to me how the curves for GAP/SOAP shown in Figure 3 were obtained. Did you obtain the data from a paper? Did you train a GAP/SOAP model using the training set you prepared for BIGDML? Please clarify.

Response 2-8): *The results in Fig. 3 were done by training the GAP/SOAP model using the same datasets used for training the BIGDML for graphene.*

Action taken:

We have added the required information in the main text to make this point clear. This is in the Methods section in a new subsection "Training GAP/SOAP models".

4) The word exact is used both in the title and in the discussion. Please refrain from using that term. I suggest changing it to "highly accurate" or something similar.

Response 2-9): *We agree with the referee that our approach cannot be considered "exact" at this moment for materials, and this explains our usage of a qualifier "Towards" in the title of our paper. Our BIGDML approach would become exact only when "gold standard" calculations [e.g. at CCSD(T) or QMC level] become available for extended systems. For example, for molecules such data is available and essentially exact MD simulations for small molecules are now within reach [see our previous publication <https://www.nature.com/articles/s41467-018-06169-2>]. Since the only remaining stepping stone to "Exact MD" for materials is the availability of high-quality reference data, we would consider the title of our manuscript a reasonable way to communicate our findings.*

Reviewer #3 (Remarks to the Author):

This paper introduces a statistical learning method ("BIGDML") for the energy and atomic forces of bulk systems with symmetries, followed by a number of demonstrations on systems including bulk and 2D materials, surface absorption, and interstitial hydrogen in a metal. The paper is well-written and the ideas are clear. My main concern is about the applicability and the merit of the method. In my opinion, it is not clear whether their method is advantageous over the state-of-the-art even in a subset of situations. My reasoning is given below.

The method has two main components in addition to the features of sGDML that was introduced by the same group of authors: the first is using a minimal image convention when computing the Coulomb matrices, which then serves as the descriptors of a system. The second is exploiting space group symmetry.

These two components both introduce shortcomings, however. First and foremost, because a global descriptor is used, and the size of the descriptor scales quadratically with the number of atoms, the system size is severely restricted, to about <100 atoms. Such small sizes are clearly too small for most applications when modeling moderately complex systems.

Response 3-1): *The first part to address is the fact that the BIGDML model is not inherently limited to systems sizes <100 atoms. For materials having a large supercell that renders direct diagonalization too costly (in terms of the amount of RAM memory used), the training process can be done by using a numerical solver, as normally done for neural networks. Hence, this would allow the BIGDML to be trained on materials with hundreds of atoms (to be done in future work). Thereby, there is indeed a wide range of applications for the BIGDML model.*

The quadratic scaling of the descriptor matrix is not a problem in practice because the descriptor is of dimension $3N \times 3N$, where N is the number of atoms. Most existing descriptors (such as SOAP or neural networks) have substantially higher dimensionality. In addition, the energy coming from the Schrödinger equation is not amenable to a local atom-based projection. Our global descriptor and gradient-domain based approach ensures that all interactions in the data can be captured correctly. We consider that quadratic scaling (with a very small prefactor) is a reasonable price to pay for the demonstrated gain in accuracy and efficiency.

Action taken:

We have added a paragraph in the introduction mentioning our answers to this question.

“It is important to mention that describing materials having several hundreds of atoms or having an exceedingly large amount of symmetries still remain challenging tasks for the model. Nevertheless, these technical issues could be addressed by utilising multiscale and composite approaches as well as numerical solvers (see section III for an extended discussion).”

Added a more extended discussion in section III of the paper.

“... Additionally, the construction of models for materials containing several hundreds of atoms per supercell could be done by using numerical solvers to overcome memory limitations as it is done for training neural networks (see e.g. LeCun(2012)).”

Second, the space group symmetry can only be exploited when the system contains such symmetries, which means that only a small fraction of systems will benefit from the methodology. Moreover, whenever interesting phenomena happen, such as a breaking of a bond in graphene, or a formation of any types of crystalline defects in bulk materials, or a surface reconstruction, the symmetries no longer apply.

Response 3-2): *This is a very interesting question, which is related to how the data is distributed in descriptor space. The case in which the system changes from a crystalline structure to a structure with a vacancy or a broken bond is perfectly described by our model, given the fact that these two phases of the system are just two different domains in descriptor space connected by transition paths. This means that structures with defects will be located away from the pristine material in descriptor space. Consequently, the BIGDML method can distinguish between the two structures and their interactions can be accurately learned provided that enough relevant data is available. The symmetries in GDML/BIGDML are determined from the data. Only relevant symmetries are used.*

Action taken:

We have added a paragraph in the Discussion section where we address this question in detail.

On the other hand, the advancement that is demonstrated in the paper on a few example systems can be easily achieved using state-of-the-art methods. None of the applications in the paper couldn't have been done using pure DFT or standard ML potentials.

Response 3-3): *Here, we strongly disagree with the statement “None of the applications in the paper couldn't have been done using pure DFT or standard ML potentials”. Performing the simulations reported in this work using DFT would require a monumental amount of computational power and time. A simple example is our simulation of the benzene/graphene system. Obtaining converged results required ~300 ps of simulations using path integral molecular dynamics with 32 beads. This demands 48 million evaluations of*

energies and forces. A good accuracy in the DFT calculations in a single node with 32 processors takes ~10 seconds. Then, this simulation would take ~5500 days, thereby this is not a practical approach. Contrasting this approach, we can generate the data, train our model on a high level of electronic structure theory and perform the PIMD simulations in about a week. In the case of standard ML potentials, this would take much longer given the large amount of data required to train the models, and still they would face accuracy limitations in the learning process.

In this regard, we disagree with the statement “the advancement that is demonstrated in the paper on a few example systems can be easily achieved using state-of-the-art methods”. As mentioned in the article, the GAP/SOAP is considered one of the most accurate and widely used models. Nevertheless, constructing general models for a specific element (e.g. carbon) can take up to years of data generation, hand-picking structure configurations which results in manually biasing the model, and still only getting moderate accuracy on specific application cases. An example of the limitations of existing models was presented by Rowe et al. [PRB 97 054303 (2018)] where they demonstrate that a dedicated graphene model can improve the test errors by up to one order of magnitude. Furthermore, even the best dedicated GAP-graphene model can be reproduced by our model using only 1% of the amount of data, and is easily outperformed by adding a few more training data points (Section Results-D-1).

In the case of more complicated environments and multi-element systems, such a performance gap will certainly increase.

Training seems to be more efficient, when comparing with the GAP models that they have fitted, but at the expense of having to use the same (small) system size when training the model and performing the simulations.

Response 3-4): *Having a dedicated MLFF model for materials that can be trained efficiently with small reference training datasets, while keeping a high model accuracy, is indeed a big advantage of the BIGDML framework. The extension to larger systems can be solved by utilizing numerical solvers for training, as normally done in the case of neural networks. Ultimately, the energy obtained as a solution of the Schrödinger equation cannot be written as a sum over atoms and this leads to a very fundamental question on the limitations of “scalable” atom-based models. This challenge is dealt with by BIGDML in a rigorous way by relying on the prediction of atomic forces with a global descriptor.*

It is also not clear how BIGDML will perform when there are crystalline defects that break the space group symmetry. It is of course okay to introduce a new method with strengths and weaknesses, but in this case, how and whether the new method is advantageous over the standard ones is not clearly demonstrated from the examples.

Response 3-5): *The BIGDML model can indeed describe such phenomena. To understand why, we have to discuss how the data is distributed in descriptor space. When a system changes from a crystalline structure to a structure with a crystalline defect, it moves from a domain in descriptor space to another one that is sufficiently separated, hence it is described by our model provided that enough data is available.*

Action taken:

We have added a paragraph in the Discussion section where we analyse this issue in detail.

The authors offer some reasons why their method is better, which are not very convincing. In the introduction, the authors state that their method is better because it is a global model without the locality assumption. However, since their system sizes are small, one can just use a BPNN/GAP/DeepMD/... with an atomic cutoff radius larger than the supercell length, or a small cutoff combined with a few message passing operations in NequIP/SchNet to achieve a global model.

Response 3-6): *The referee is right in stating that many different choices and approximations need to be made when constructing a given MLFF. In the limit of large model flexibility and infinite data, many currently developed MLFFs would yield accurate results. In contrast, most MLFF are built with small data and fixed*

model flexibility. This often yields quite different performance in actual applications. For example, recent work highlights very large differences in performance between kernels and neural networks for molecular dynamics datasets (<https://pubs.rsc.org/en/content/articlelanding/2021/SC/D1SC03564A>). The BIGDML framework relies on well-established physical principles, employs all known symmetries of periodic systems and uses the simplest possible descriptor (Coulomb matrix) without the need of atom prototyping or localization of interactions. Hence, we consider our approach "the right tool" (accurate, computationally straightforward, and data efficient) for modelling extended materials with moderately-sized supercells. Our empirical results and comparison with alternative kernel-based approaches fully confirm the reasoning above.

Action taken:

This discussion is already present in the text, but we have extended it to include the above-mentioned argument in the Discussion section.

Another reason is that "additional advantage of a global representation is that cross-correlations between forces on different atomic species are dealt with rigorously, at variance with existing atomic representations." But this is not clear to me and should be expanded.

Response 3-7): *The global description of the system used by the BIGDML model treats with equal footing all the interatomic pairs. This is due to the explicit use of a global force covariance, which allows capturing correlations between all-atom configurations and many-body atomic forces within a given material's supercell structure. Thereby, learning the relevant dominant and subtle interatomic interactions at different spatial scales.*

Action taken:

*This discussion has been added to the Introduction of the paper.
"Specifically, MLFFs based on the locality approximation construct separate models for each atom type. In contrast, the BIGDML model employs a global force covariance, allowing many-body correlations between atomic forces in a given supercell structure and capturing relevant interatomic interactions at different spatial scales."*

They say that "Our choice of CM with PBC enforced using MIC is the simplest and most efficient choice", but it is not clear why their choice is better than the Sine matrix or CM combined with an Ewald-sum.

Response 3-8): *The use of a CM+MIC simple representation that provides smooth local interactions and a slow decay of interatomic interactions ($1/|r_i - r_k|$) to capture long distance correlations. Regarding the CM+Ewald-sum, even the authors acknowledge in the original paper [Int. J. Quantum Chem.115, 1094 (2015)] that is not a suitable representation given that is not a closed form and converges very slowly with the supercell copies considered, hence it is not a good representation. In the case of the Sine matrix, this is a good representation for studying the crystal structure space of materials at equilibrium, but it is not a valid descriptor for MD data given that its main feature (i.e. projection of the full supercell to the unit cell) can not provide an accurate measure of two atoms in different cells of the supercell since, as stated in the original paper, "only depends on the positions of the atoms in a single unit cell". Now, in the case of a taking the supercell of the system as a unit cell in the Sine matrix, we get a descriptor whose elements ij are $\sim 1/\sin^2(|r_i - r_j|)$ which is an extremely local descriptor, hence less expressive (See next Figure).*

This discussion has been added in section II-A.

I also have some doubts about the novelty of the method itself. The minimal image convention is used through and through in any simulation studies involving periodic boundary conditions. The exploitation of the discrete symmetry is equivalent to data augmentation, which has been widely used.

Response 3-9): *In our paper, we present the combination of these features as a whole and its mathematical encoding in the BIGDML method, which at the core is responsible for the robustness of the model. It is not the purpose of the paper to claim that the novelty is in the separate use of these elements, but instead to highlight the great advantage of combining the full symmetry group of a crystal lattice and a global representation of the system in an ML setting for describing materials dynamics.*

Minor points:

It is probably much clearer to draw the learning curves using the log scale on the x-axis.

Response 3-10): *We thank the reviewer for the suggestion.*

Action taken:

We have modified both the x-axis and the y-axis in Figs. 2 and 3 to log scale.

More information on how the GAP/SOAP models were fitted should be given. It is not clear whether reasonable GAP parameters were chosen for these fits.

Response 3-11): *We thank the reviewer for highlighting this issue.*

Action taken:

We have extended this explanation in Methods section in a new subsection "Training GAP/SOAP models".

The acronym "BIGDML" is misleading as it hints at large-scale ML models but in fact both the training points and the typical system sizes are small.

Response 3-12): *This acronym was chosen based on the features the method incorporates (Bravais-Inspired Gradient Domain Machine Learning), as well as to hint that the model is dedicated to extended (periodic) systems.*

Reviewers' comments:

Reviewer #2 (Remarks to the Author):

This manuscript received three independent critical reviews and all referees agreed that the method has limitations that might prevent general applicability of the approach. The authors have replied to these comments arguing that these limitations can be easily overcome yet provide no evidence that this is the case. For instance, there are still no applications to larger systems nor to phase transitions nor to the liquid state.

As I mentioned in my previous review, the fact that the method is not perfect does not prevent publication. In my opinion the issue that does prevent publication of this manuscript is that the achievements of the method are exaggerated. The method does show good performance but only in very specific cases. In contrast, it is presented as if this performance applied generally. I don't believe this manuscript presents real progress towards more accurate general force fields. If the authors had trained a model able to describe many different phases, including the liquid, and had shown that they can do it with 200 configurations and achieving errors substantially below 1 meV per atom, I would have endorsed the publication of this manuscript. For the reasons detailed above, my recommendation is to reject this manuscript.

On the other hand, I congratulate the authors for following a different path compared to the rest of the scientific community. I think their method has tremendous potential but from the applications presented here it is too early to judge if it will be really useful. I also think the quality of the presentation of the manuscript is very good, the level of execution is high, and the research described here definitely has a place in the literature. However, in my opinion, a high profile journal such as Nature Communications is not appropriate in this case, especially when claims are exaggerated.

Reviewer #3 (Remarks to the Author):

All three referees spotted the main problems of the manuscript: 1) poor scalability, 2) not transferable between systems of different sizes, and 3) the applicability to other crystalline structures or crystals with defects is not clear. In response to these comments, the authors mainly added some claims, e.g. "... Additionally, the construction of models for materials containing several hundreds of atoms per

supercell could be done by using numerical solvers to overcome memory limitations as it is done for training neural networks (see e.g. LeCun(2012)).}."there is no fundamental limitation in the BIGDML model to describe them. Two different crystalline phases in descriptor space are simply two domain regions (each having its own set of symmetries) separated by one or more transition paths."

To me, these are basically conjectures that say nothing more than "the problem can in principle be solved", without any convincing demonstration. I suggest that the authors actually demonstrate the scalability and the transferability of their methods.

REPLY TO REVIEWERS COMMENTS

First, we would like to thank both reviewers for carefully reading and analyzing our manuscript and for the suggestions to improve the article. Below, you'll find a point-by-point reply to the remarks.

From the comments, two main concerns were raised, scalability and transferability of the BIGDML.

In the latest version of the manuscript we have addressed the scalability issue by adding a larger multi-element system, a perovskite material containing 160 atoms. This application not only shows that systems with larger supercells are within the description capabilities of the BIGDML, but also highlights the great advantage of using a global model when describing fluxional multi-element materials. Here, state-of-the-art results were obtained even for 100 training samples. Furthermore, in order to show that scalability on the training side is not a limitation, we have trained this systems with upto 1000 training samples.

The BIGDML model offers the unique possibility of truly learning global interactions, meaning that it accurately captures local and delocalized interactions without the need of using non-unique energy partitioning schemes or other approximations. Such a prominent feature inhibits knowledge transfer between different systems. This means that in order to create a transferable model, a localization of the interactions is mandatory. Hence, we acknowledge that, in its current form, our model is not capable of exploiting knowledge from a learned model and transferring it to a new system. Then, in the DISCUSSION section (page 12) we provide some mathematical ideas where we discuss different ways to address the transferability problem. In particular we discuss a construction or convolutional approach and a deconstruction of the global features in the model. This gives an overview of how the BIGDML framework is envisioned as a base model to further develop towards constructing more general and efficient force fields for materials modeling.

The rest of the points raised by the reviewers are addressed point-by-point in the following part.

Reviewer #2 (Remarks to the Author):

This manuscript received three independent critical reviews and all referees agreed that the method has limitations that might prevent general applicability of the approach. The authors have replied to these comments arguing that these limitations can be easily overcome yet provide no evidence that this is the case. For instance, there are still no applications to larger systems nor to phase transitions nor to the liquid state.

Response #2-1): *We would like to remark that the developed BIGDML framework is the first approach for constructing global machine-learned force fields for materials, which naturally enables the possibility of describing complex non local interactions on materials. Now, even though the mathematical formulation of our proposal presents no inherent theoretical limitations to describe general systems, we agree with the reviewer that in its current state the BIGDML is not yet applicable to highly diffusive systems. For this reason, we have softened our claims regarding the description of phase transitions while providing arguments of which is the development direction to address this task (This has been added in the DISCUSSION section).*

Then, we focused on the strengths that the model currently has. For example, we have included a new system, a CsPbBr₃ perovskite containing 160 atoms per supercell. The discussion about this new system is shown in the section RESULTS-D-1 and its learning curves are shown in Fig. 4 of the manuscript. Such multi-element and highly fluxional material pose a challenging learning task, nevertheless the BIGDML manages to reach energy errors of 1 meV/atom even with a training set of 100 data points. Furthermore, to demonstrate that our model can handle large training data sets for systems with over 160 atoms, we have trained models with up to 1000 data points. Hence, with this example we show that the BIGDML can describe the plethora of materials of interest containing up to 160 atoms with high accuracy and addresses the scalability comments by the reviewers.

As I mentioned in my previous review, the fact that the method is not perfect does not prevent publication. In my opinion the issue that does prevent publication of this manuscript is that the achievements of the method are exaggerated. The method does show good performance but only in very specific cases. In contrast, it is presented as if this performance applied generally.

Response #2-2): *As stated by the reviewer, our model shows good performance on the systems under consideration. Thereby, we would like to highlight that a similarly good performance is expected to be obtained when applying our methodology to systems belonging to the same family of materials shown in our manuscript.*

Nevertheless, we agree with the reviewer that we didn't emphasize enough the type of systems our model can describe, hence it could be read as claiming generality. To address this issue, we have explicitly added a paragraph in section RESULTS-D discussing which family of materials each one of the applications represent and why we expect such results. For example, from Fig. 2 we expect that in general, given the performance of the trained models for Na, Au, and Pd, mono-atomic materials with cubic unit cells will be accurately described by the BIGDML. Another example is given by the newly incorporated CsPbBr₃ perovskite containing 160 atoms, which shows that the model can handle and accurately learn large multi-element and fluxional materials. Then, a similar performance is expected when applied to a family of materials such as similar perovskites or materials with similar structural characteristics. Hence, this will better picture the range of applicability of the model.

I don't believe this manuscript presents real progress towards more accurate general force fields. If the authors had trained a model able to describe many different phases, including the liquid, and had shown that they can do it with 200 configurations and achieving errors substantially below 1 meV per atom, I would have endorsed the publication of this manuscript. For the reasons detailed above, my recommendation is to reject this manuscript.

Response #2-3): *We agree with the reviewer that in its current state the BIGDML is not yet applicable to highly diffusive systems, and considering the reviewers' comments we have softened our claims in this regard, and focused on the strengths that it currently has.*

On the other hand, we find the reviewer's request to model large systems (including liquids and phase transitions) unreasonable for the current state of the art of machine learning force fields (MLFF). Surely, many seminal force fields exist that are being used to model phase transitions and liquids. But all of them are based on rather simplified localized interactions, neglecting many important symmetries and non-local effects. We would like to stress that BIGDML is the first model that is able to convincingly address all of these basic requirements and obtain sub-meV accuracy per atom in MLFFs for materials. This achievement is demonstrated in our paper for a wide range of applications to materials including pristine and defect-containing 2D and 3D semiconductors and metals, as well as chemisorbed and physisorbed atomic and molecular adsorbates on surfaces, all this with accuracy and efficiency that exceeds the state of the art by at least an order of magnitude. Furthermore, to address the most pressing issues brought up by the referees, we have included a new system, a CsPbBr₃ perovskite containing 160 atoms per supercell. This addresses the reviewer's complaint on the scalability to larger systems and more challenging crystalline structures. This new analysis is shown in the section RESULTS-D-1 and its learning curves are shown in Fig. 4 of the manuscript. Such multi-element and highly fluxional material pose a challenging learning task, nevertheless the BIGDML manages to reach energy errors of 1 meV/atom even with a training set of 100 data points. Furthermore, to demonstrate that our model can handle large training data sets for systems with over 160 atoms, we have trained models with up to 1000 data points. Hence, with this example we show that the BIGDML can describe the plethora of materials of interest containing up to 160 atoms with high accuracy and addresses the scalability comments by the reviewers.

On the other hand, I congratulate the authors for following a different path compared to the rest of the scientific community. I think their method has tremendous potential but from the applications presented here it is too early to judge if it will be really useful. I also think the quality of the presentation of the manuscript is very good, the level of execution is high, and the research described here definitely has a place in the literature. However, in my opinion, a high profile journal such as Nature Communications is not appropriate in this case, especially when claims are exaggerated.

Response #2-4): *Here Referee #2 agrees that our approach is following a "different path compared to the rest of the scientific community" and that "their method has tremendous potential". We are convinced that these two aspects already make our paper deserving to be published in Nature Communications. Obviously, a method that deviates strongly from the status quo will be evaluated very critically by reviewers, and this is clearly the case with our manuscript. We hope that the additional discussions incorporated in the text, where we have softened our claims and presented extensive discussions regarding the needed changes in the model to address the limitations raised by the*

referees, and the addition of a multi-element highly-fluxional perovskite system with 160 atoms per unit cell provides convincing evidence for the scalability of the BIGDML framework, are enough evidence to the potential of our model.

Reviewer #3 (Remarks to the Author):

All three referees spotted the main problems of the manuscript: 1) poor scalability, 2) not transferable between systems of different sizes, and 3) the applicability to other crystalline structures or crystals with defects is not clear. In response to these comments, the authors mainly added some claims, e.g. "... Additionally, the construction of models for materials containing several hundreds of atoms per supercell could be done by using numerical solvers to overcome memory limitations as it is done for training neural networks (see e.g. LeCun(2012)).}" "there is no fundamental limitation in the BIGDML model to describe them. Two different crystalline phases in descriptor space are simply two domain regions (each having its own set of symmetries) separated by one or more transition paths."

To me, these are basically conjectures that say nothing more than "the problem can in principle be solved", without any convincing demonstration. I suggest that the authors actually demonstrate the scalability and the transferability of their methods.

Response #3-1): *We would like to thank the reviewer for carefully analyzing our manuscript and for the suggestions to improve the article. We have softened our claims and focused on the strengths that BIGDML currently has. For example, we would like to remark that the BIGDML framework is the first approach for constructing global machine-learned force fields for materials, properly handling non-local interactions in complex materials. Furthermore, to address the most pressing issues brought up by the referee, we have included a new system, a CsPbBr₃ perovskite containing 160 atoms per supercell. This addresses the reviewer's complaint on the scalability to larger systems and more challenging crystalline structures. This new analysis is shown in the section RESULTS-D-1 and its learning curves are shown in Fig. 4 of the manuscript. Such multi-element and highly fluxional material pose a challenging learning task, nevertheless the BIGDML manages to reach energy errors of 1 meV/atom even with a training set of 100 data points. Furthermore, to demonstrate that our model can handle large training data sets for systems with over 160 atoms, we have trained models with up to 1000 data points. Hence, with this example we show that the BIGDML can describe the plethora of materials of interest containing up to 160 atoms with high accuracy and addresses the scalability comments by the reviewers.*

Already in the first round of reviews, the referees clearly acknowledged the potential of BIGDML. For example, the referees highlighted the robustness of the method and the fact that the article provides all the information necessary to reproduce the reported results. Hence, with the new evidence from the extended analysis, and the extensive study previously incorporated in the manuscript, and with the very valuable contributions from the referees, we are convinced that our article has the merits to be published in Nature Communications.